# Copy number load predicts outcome of metastatic colorectal cancer patients receiving bevacizumab combination therapy

Dominiek Smeets et al.[#]

Increased copy number alterations (CNAs) indicative of chromosomal instability (CIN) have been associated with poor cancer outcome. Here, we study CNAs as potential biomarkers of bevacizumab (BVZ) response in metastatic colorectal cancer (mCRC). We cluster 409 mCRCs in three subclusters characterized by different degrees of CIN. Tumors belonging to intermediate-to-high instability clusters have improved outcome following chemotherapy plus BVZ versus chemotherapy alone. In contrast, low instability tumors, which amongst others consist of *POLE*-mutated and microsatellite-instable tumors, derive no further benefit from BVZ. This is confirmed in 81 mCRC tumors from the phase 2 MoMa study involving BVZ. CNA clusters overlap with CRC consensus molecular subtypes (CMS); CMS2/4 xenografts correspond to intermediate-to-high instability clusters and respond to FOLFOX chemotherapy plus mouse avastin (B20), while CMS1/3 xenografts match with low instability clusters and fail to respond. Overall, we identify copy number load as a novel potential predictive biomarker of BVZ combination therapy.

Colorectal cancer (CRC) is the third most commonly diagnosed malignancy in both men and women and is associated with high mortality and morbidity[1]. Almost half of patients diagnosed with CRC develop metastatic disease (mCRC). Current treatment for RAS mutant mCRC includes 5-fluoruracil-based standard of care chemotherapy (e.g., monotherapy, XELOX/FOLFOX, FOLFIRI, and FOLFOXIRI) combined with the angiogenesis inhibitor bevacizumab (BVZ). Results from phase III clinical trials have indeed demonstrated that the addition of BVZ to chemotherapy improves response rate and prolongs survival of mCRC patients[2,3]. Nevertheless, only a subset of patients respond, and overall clinical benefit of BVZ is limited with most patients ultimately succumbing[4]. Moreover, BVZ therapy is associated with a specific side effect profile and high treatment costs. Although we[5–7] and others[8–13] have previously proposed several novel genomic entities as putative BVZ response predictors, to date no robust validated biomarker for BVZ in CRC has emerged. Thus, understanding BVZ resistance mechanism(s) and identifying unambiguous biomarkers to predict patient outcome remain clinically relevant questions.

To address these issues, we have drawn on knowledge emerging from recent efforts to characterize the complex genomic alterations that underpin CRC aetiology[14,15]. Arguably, the most comprehensive studies have emerged from The Cancer Genome Atlas (TCGA), which has identified multiple driver genes and CRC genetic phenotypes including hypermutators (12% MSI; 5% POLE/POLD1), chromosomal instability [CIN] (65–70%) and CpG island methylator (15%)[14,15]. This led to the identification of novel biomarkers to predict response to targeted therapies, such as KRAS for anti-EGFR therapy[16], and more recently, high tumor mutational burden for anti-PD1/PDL1 checkpoint immunotherapy[17]. Additional studies have focused on reclassifying CRC based on tumor expression data, resulting in a new Consensus Molecular Subtype (CMS) classification system of CRC, which is now poised to significantly impact future clinical stratification and CRC subtype-based targeted intervention[18–24].

In this study, the ANGIOPREDICT (APD) consortium (www.angiopredict.com) studies chromosomal instability (CIN) and its impact on treatment outcome in mCRC. Specifically, we explore how tumors cluster based on the genome-wide distribution of copy number alterations (CNAs) and define 3 CNA clusters. We correlate each of these clusters with tumor and clinical characteristics, tumor mutation burden, CMS subtypes and treatment outcome. We show that tumors belonging to clusters with intermediate-to-high instability have improved outcome after BVZ combination therapy, whereas tumors characterized by low instability derive no further benefit from BVZ. Finally, we also functionally confirm our findings in mouse xenografts. All study characteristics and findings are reported according to REMARK criteria[25].

| **Table 1 Clinical info** | | | | | | |
|---|---|---|---|---|---|---|
| | **BVZ** | | **SOC** | | **MOMA** | |
| | $n = 185$ | % | $n = 224$ | % | $n = 81$ | % |
| *Gender* | | | | | | |
| Female | 72 | 38.9 | 84 | 37.5 | 31 | 39 |
| Male | 113 | 61.1 | 140 | 62.5 | 50 | 61 |
| *Age (years)* | | | | | | |
| >65 | 79 | 42.7 | 100 | 44.6 | 21 | 26 |
| ≤65 | 103 | 55.7 | 124 | 55.4 | 60 | 74 |
| Missing values | 3 | 1.6 | 0 | 0.0 | 0 | 0 |
| *T-classification* | | | | | | |
| 1 | 2 | 1.1 | 0 | 0.0 | 1 | 1 |
| 2 | 16 | 8.6 | 12 | 5.4 | 3 | 4 |
| 3 | 120 | 64.9 | 155 | 69.2 | 29 | 36 |
| 4 | 41 | 22.2 | 52 | 23.2 | 15 | 18 |
| Missing values | 6 | 3.2 | 5 | 2.2 | 33 | 41 |
| *N-classification* | | | | | | |
| 0 | 41 | 22.2 | 67 | 29.9 | 8 | 10 |
| 1 | 68 | 36.8 | 74 | 33.0 | 17 | 21 |
| 2 | 63 | 34.1 | 72 | 32.1 | 22 | 27 |
| Missing values | 13 | 7.0 | 11 | 4.9 | 34 | 42 |
| *KRAS* | | | | | | |
| wt | 92 | 49.7 | 7 | 3.1 | 33 | 41 |
| mut | 48 | 25.9 | 2 | 0.9 | 46 | 57 |
| Missing values | 45 | 24.3 | 215 | 96.0 | 2 | 2 |
| *BRAF* | | | | | | |
| wt | 110 | 59.5 | 7 | 3.1 | 72 | 89 |
| mut | 16 | 8.6 | 0 | 0.0 | 7 | 9 |
| Missing values | 59 | 31.9 | 217 | 96.9 | 2 | 1 |
| *BVZ* | | | | | | |
| Yes | 185 | 100.0 | 0 | 0.0 | 81 | 100 |
| No | 0 | 0.0 | 224 | 100.0 | 0 | 0 |
| *Backbone* | | | | | | |
| FP | 12 | 6.5 | 102 | 45.5 | 0 | 0 |
| FP-OX | 136 | 73.5 | 16 | 7.1 | 0 | 0 |
| FP-IRI | 37 | 20.0 | 106 | 47.3 | 0 | 0 |
| FP-OX-IRI | 0 | 0.0 | 0 | 0.0 | 81 | 0 |
| Total | 185 | 100 | 224 | 100 | 81 | 100 |

Summary of clinical info for mCRC patients receiving either BVZ (185 APD patients) or standard-of-care chemotherapy (19 APD, 205 CAIRO patients) and the MoMa clinical trial SOC: standard-of-care, wt: wild-type, mut: mutated, BVZ: bevacizumab, FP: fluoropyrimidin, IRI: irinotecan, OX: oxaliplatin

## Results

**Study population.** Within APD, tumor biopsies and clinical data were retrospectively collected from 274 mCRC patients. High-quality low-coverage whole-genome sequencing (shallow-seq) data obtained for 215 of these samples were reported previously (Supplementary Table 1)[26]. Additionally, we performed whole-exome sequencing (WES) on 156 samples with paired germ-line and tumor DNA available. The average coverage was 59.6x with a standard deviation of 43.8×, and 88.3 ± 9.7% of the exome was sequenced with >10x coverage (Supplementary Note 1, Supplementary Data 1).

195 out of 215 patients received a treatment involving BVZ. Specifically, patients received BVZ combined with a fluoropyrimidine (FP) chemotherapy backbone, either alone ($n = 12$) or in combination with either irinotecan (IRI) or oxaliplatin (OX) ($n = 173$). Ten patients were excluded because they received either BVZ monotherapy or another combination therapy, or because treatment data was missing. A small number of patients received BVZ in 2nd or even later lines ($n = 13$ and 6, respectively). Since there were no survival differences, these patients were retained for further analysis (Supplementary Table 2). Patient characteristics of the resulting 185 BVZ-treated APD patients are summarized in Table 1. Twenty patients did not receive BVZ, of which one patient was treated with FP, OX and cetuximab and was therefore excluded.

Additionally, we obtained publicly available CNA data from 205 mCRC patients included in the CAIRO phase 3 trial (NCT00312000) randomized for FP and IR ($n = 104$) versus FP alone ($n = 101$) (Supplementary Table 1). CNA data for these mCRC tumors were generated using Agilent oligonucleotide

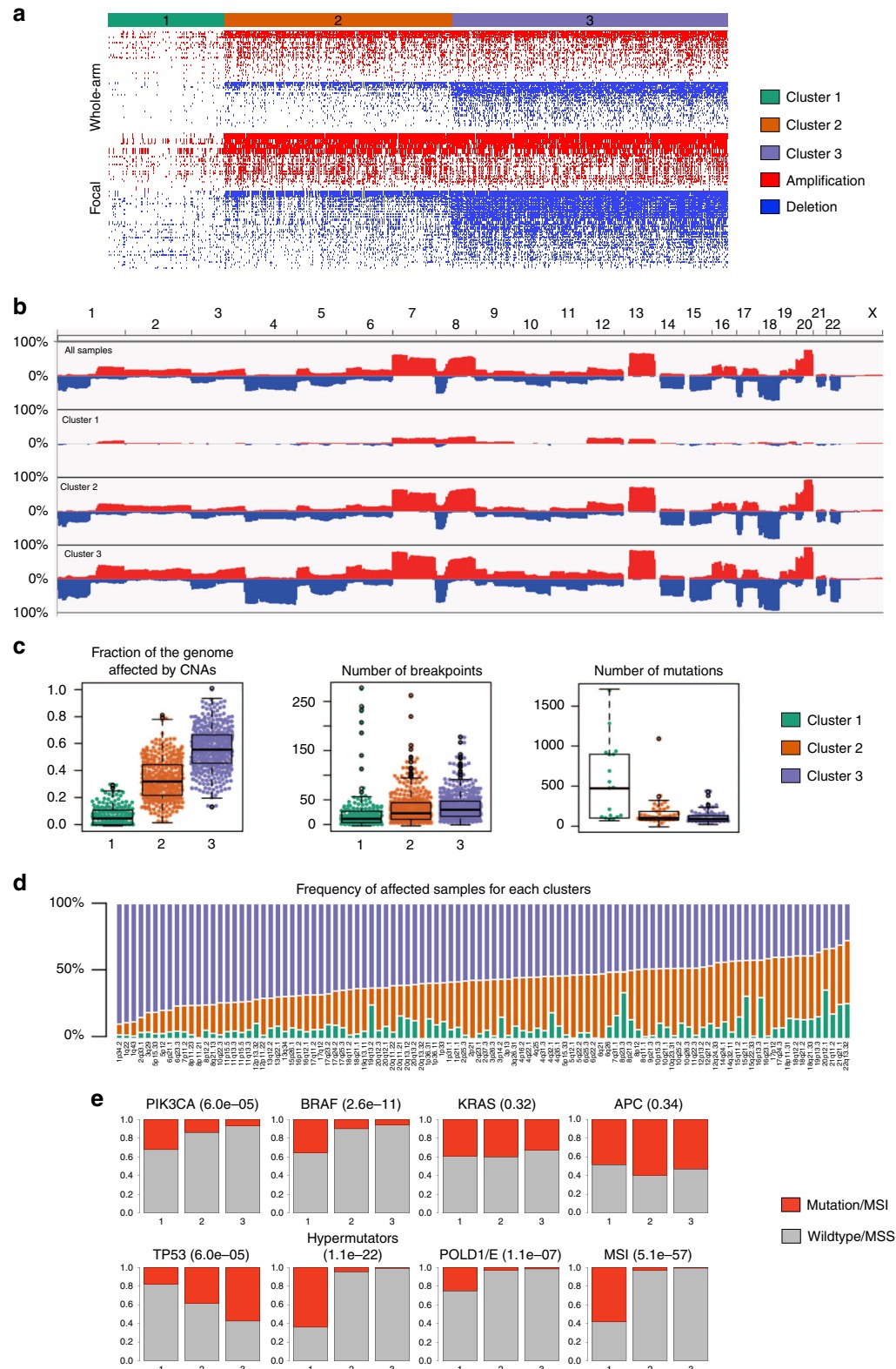

hybridization arrays[13]. No WES data were available. We also accessed CNA data for 499 CRC patients from TCGA (http://gdac.broadinstitute.org/), a minority ($n = 63$) being mCRC patients (Supplementary Table 1). For 152 patients included in TCGA, both WES and CNA data were available. Survival data, gender, grade, age and stage distributions across these 3 different cohorts are summarized in Supplementary Figure 1.

**Unsupervised clustering of CNAs reveals 3 consensus clusters**. First, we applied GISTIC[27] on all tumors for which CNA data were available to identify recurrent CNAs (FDR < 0.05). This analysis revealed 43 recurrent focal amplifications and 59 recurrent focal deletions, as well as several whole-arm aberrations (Supplementary Data 2, Supplementary Figure 2). Most of these CNAs were also detected in the individual cohorts

**Fig. 1** Clustering of primary and metastatic colorectal cancer. **a** Unsupervised hierarchical clustering of copy number profiles of primary and metastatic CRC ($n = 908$) tumors into 3 consensus CNA subgroups (termed CNA clusters 1, 2, and 3) based on recurrent CNAs as determined by GISTIC. Presence of recurrent amplifications (red) and deletions (blue) for each sample is shown. The 908 tumors represent 204 APD, 205 CAIRO and 499 TCGA tumors for which copy number data were available. **b** IGV plot showing how frequent each of the chromosomal regions (Y-axis) is affected by amplifications (red) or deletions (blue) in tumors belonging to CNA cluster 1, 2 and 3. **c** Genomic characterization of the 3 clusters for: the fraction of the genome affected by CNAs, the number of breakpoints and the number of mutations. Box plots show the median, the 25th and 75th percentiles, Tukey whiskers (median ± 1.5 times interquartile range). **d** Frequency of affected samples per cluster for each of the 102 significant amplifications or deletions (X-axis). **e** Distributions of the mutation frequency of *PIK3CA, BRAF, KRAS, APC, TP53, POLD1/POLE*, hypermutators, and MSI status for each cluster. The presence of a mutation or positive status for MSI or hypermutator is depicted in red and absence in grey. Fisher P-values are indicated between parentheses

(Supplementary Figure 3). Next, we performed unsupervised hierarchical consensus clustering in an iterative manner based on CNA status of these 102 focal and 39 whole-arm CNAs. This analysis further revealed 3 clusters to which patients could be assigned (Fig. 1a; Supplementary Figure 4). Particularly, clusters 1, 2 and 3 consisted of 170 (18.7%), 334 (36.8%), and 404 (44.5%) tumors, respectively.

Characterization of CNA load in these clusters revealed that with increasing cluster number, an increasing number of chromosomal breakpoints and a higher proportion of the genome affected by CNAs (i.e., high copy number load) were detected (Fig. 1b). Particularly, cluster 1 showed almost no CNAs or breakpoints, while cluster 3 exhibited the highest number of CNAs or breakpoints ($P < 0.001$, Student's t-test) (Fig. 1c). A small number of CNAs were, however, more frequent in CNA cluster 1 or 2 compared to cluster 3, indicating that cluster membership was not only determined by copy number load (Fig. 1d). Neither the number of chromosomal breakpoints nor proportion of the genome affected by CNAs depended on tumor percentage (Supplementary Figure 5).

Cluster 1 was enriched for tumors with high mutational burden, including tumors of the MSI subtype ($P = 4.1 \times 10^{-56}$, Fisher's exact test) and tumors with *POLE* and *POLD1* mutations ($P = 9.1 \times 10^{-07}$ Fisher's exact test). Tumors in cluster 1 were also enriched for *BRAF* ($P = 2.1 \times 10^{-10}$, Fisher's exact test) and *PIK3CA* ($P = 4.8 \times 10^{-4}$, Fisher's exact test) mutations, while *TP53* mutations were more frequent in clusters 2 and 3 ($P = 7.2 \times 10^{-6}$, Fisher's exact test). Notably, *APC* and *KRAS* mutations were evenly distributed among all 3 clusters, consistent with their early genetic role in CRC development (Fig. 1d). Next, we performed survival analyses to assess the prognostic relevance of the clusters (Fig. 2a). Treatment data were not considered as they were not available for most TCGA samples. Multivariate analysis using a Cox regression correcting for clinical covariates (gender, age and stage) revealed that none of the 3 clusters significantly contributed to prognosis. However, clusters 2 and 3 were significantly enriched amongst tumors with high regional lymph node involvement ($P = 3.4 \times 10^{-7}$, chi-square test), higher stage ($P = 1.5 \times 10^{-12}$, chi-square test) and distant metastasis ($3.2 \times 10^{-9}$, chi-square test) (Fig. 2b), suggesting that high copy number load tumors were more frequent in mCRC.

**Clinical and genomic characteristics of CNA clusters in mCRC.** Subsequently, we selected only those tumors collected from patients with metastatic CRC and employed GISTIC to re-cluster based on CNAs ($n = 124$ from APD, $n = 205$ from CAIRO, and $n = 63$ from TCGA). 80 out of 204 APD biopsies were collected at the time of resection of an early stage CRC (for which they later developed metastatic relapse). For the latter patients, we considered tumor characteristics collected at the time of resection, but treatment and survival data at metastatic relapse. Therefore, these 80 CRCs were not included in the current mCRC cluster analysis.

Clustering of CNA data from mCRC identified 3 CNA clusters, with up to 91.5% of tumors belonging to the same cluster and 7.2% of tumors switching between clusters 2 and 3. Cluster 1 (10.0% of tumors) exhibited almost no CNAs and was enriched for hypermutated tumors ($P = 5.6 \times 10^{-5}$, Fisher's exact test), *BRAF* ($P = 0.07$, Fisher's exact test) and *PIK3CA* mutations ($P = 0.015$, Fisher's exact test) (Supplementary Figure 6), whereas cluster 2 and 3 (39.8 and 50.2% of patients, respectively) exhibited an increasing number of CNAs and was enriched for *TP53* mutations. Fewer samples clustered to cluster 1, possibly because MSI tumors are less common in the metastatic setting. Indeed, the proportion of CRC samples belonging to cluster 1 was 18.7% for all CRCs versus 10.2% for mCRCs (Supplementary Figure 4). Furthermore, since the distribution of samples from each cohort over the 3 clusters was comparable, clustering was independent from the technology used to detect CNAs (Supplementary Figure 4).

**Patients in CNA cluster 2 and 3 benefit from BVZ.** Next, we compared cluster membership, copy number load and survival between the 80 APD tumors collected at early stage CRC and the 124 tumors collected from metastatic disease. We failed to observe differences between both groups (Supplementary Figure 7), and therefore pooled the 204 tumors from APD with the 205 tumors from CAIRO to assess effects of CNA cluster membership on treatment outcome. Overall, this resulted in 185 mCRCs receiving BVZ combined with chemotherapy and 224 mCRCs (19 from APD and 205 from CAIRO) receiving chemotherapy alone (Table 1). Multivariate Cox regression revealed that both CNA cluster 2 and 3 correlated with improved progression-free survial (PFS). Particularly, relative to cluster 1, hazard ratios (HRs) for cluster 2 and 3 were: 0.48 (CI 0.33–0.70; $P < 0.001$, Cox regression) and 0.57 (CI 0.39–0.83; $P = 0.003$, Cox regression) (Fig. 2c, d). Notably, besides age, stage and gender, this analysis was also corrected for chemotherapy backbone and BVZ treatment, with BVZ (HR = 0.70, $P = 0.0042$, Cox regression), and to a lesser extent also chemotherapy backbone (HR = 0.79, $P = 0.082$, Cox regression), significantly affecting survival.

Next, we assessed whether each CNA cluster similarly affected PFS in response to BVZ treatment. Multivariate analysis only in patients receiving BVZ ($n = 185$) revealed that tumors belonging to cluster 2 and 3 responded better to BVZ: HRs relative to cluster 1 were 0.24 ($P = 1.11 \times 10^{-5}$, CI 0.12–0.45, Cox regression) and 0.27 ($P = 2.48 \times 10^{-5}$, CI 0.14–0.49, Cox regression) for tumors belonging to clusters 2 and 3, respectively. Inclusion of an interaction term between CNA cluster membership and BVZ treatment was further significant for cluster 2 and 3 ($P = 0.040$ and $P = 0.0108$), but as expected not for cluster 1. Likewise, at the level of OS, HRs were 0.46 and 0.35 ($P = 2.93 \times 10^{-2}$, CI 0.23–0.92 and $P = 2.61 \times 10^{-3}$, CI 0.18–0.70, Cox regression) (Fig. 3a, b). In non-BVZ treated patients ($n = 224$), HRs were not significant for cluster 3 (HR = 0.72, $P = 0.18$, CI 0.45–1.16 for PFS and HR = 0.84, CI 0.49–1.44 for OS, Cox regression), while for cluster 2 patients, a borderline significant effect was observed for PFS (HR = 0.57, $P = 2.4 \times 10^{-2}$, CI 0.35–0.93, Cox regression), which was not confirmed at the OS level (Supplementary Figure 8). The effect of BVZ did not depend on whether the biopsy was collected at resection for an earlier CRC or at the time

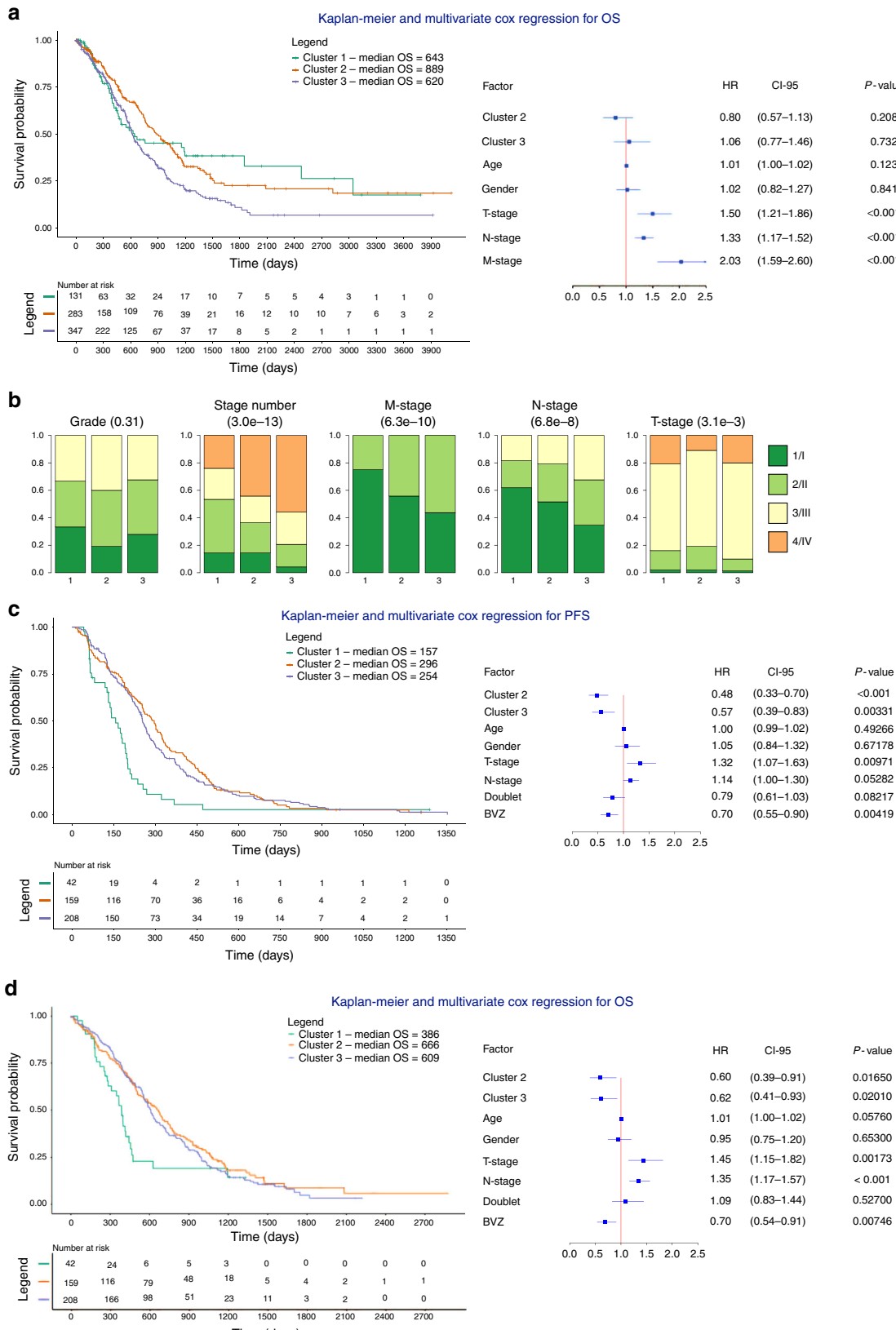

of metastatic disease (Supplementary Figure 9). Overall, these data indicate that CNA clusters have only modest prognostic effects but display significant BVZ-associated predictive effects.

We also stratified our cohort based on CNA cluster and compared survival between BVZ and non-BVZ treated patients. Tumors belonging to cluster 2 or 3 showed improved response to

BVZ with HRs of 0.58 and 0.69 respectively ($P = 9.80 \times 10^{-3}$, CI 0.38–0.88 and $P = 3.05 \times 10^{-2}$, CI 0.50–0.97, Cox regression). Strikingly, BVZ did not prolong survival in cluster 1 tumors (HR $= 1.15$, $P = 0.71$, CI 0.55–2.41, Cox regression, Fig. 4a–c). The effect of BVZ was therapeutically relevant, as patients in cluster 2 and 3 were characterized by a longer median survival of

**Fig. 2** Multivariate Cox regression and clinical characteristics of CNA clusters. **a** Kaplan-Meier plots and multivariate Cox regression with hazard ratios, 95% confidence intervals and P-values for CNA clusters are shown while correcting for the relevant covariates in all ($n = 908$) CRC samples. Cluster 1 is considered a reference. There is no difference for cluster identity, instead T-stage, N-stage, and M-stage are significant covariates in the model. **b** Clinical characterization of the CNA clusters. Clusters 2 and 3 are enriched for tumors with high T-stage, N-stage, and M-stage. Chi-squared P-values are presented between parentheses. **c**, **d** Kaplan-Meier plots and multivariate Cox regression with hazard ratios, 95% confidence intervals and P-values for CNA clusters are shown while correcting for the relevant covariates in mCRC samples treated ± BVZ ($n = 409$). Cluster 1 is considered a reference. Doublet stands for mono-chemotherapy (FP) or a combination of chemotherapy (FP-OX, FP-IRI). Clusters 2 and 3 are correlated with better PFS and OS independent of the other covariates

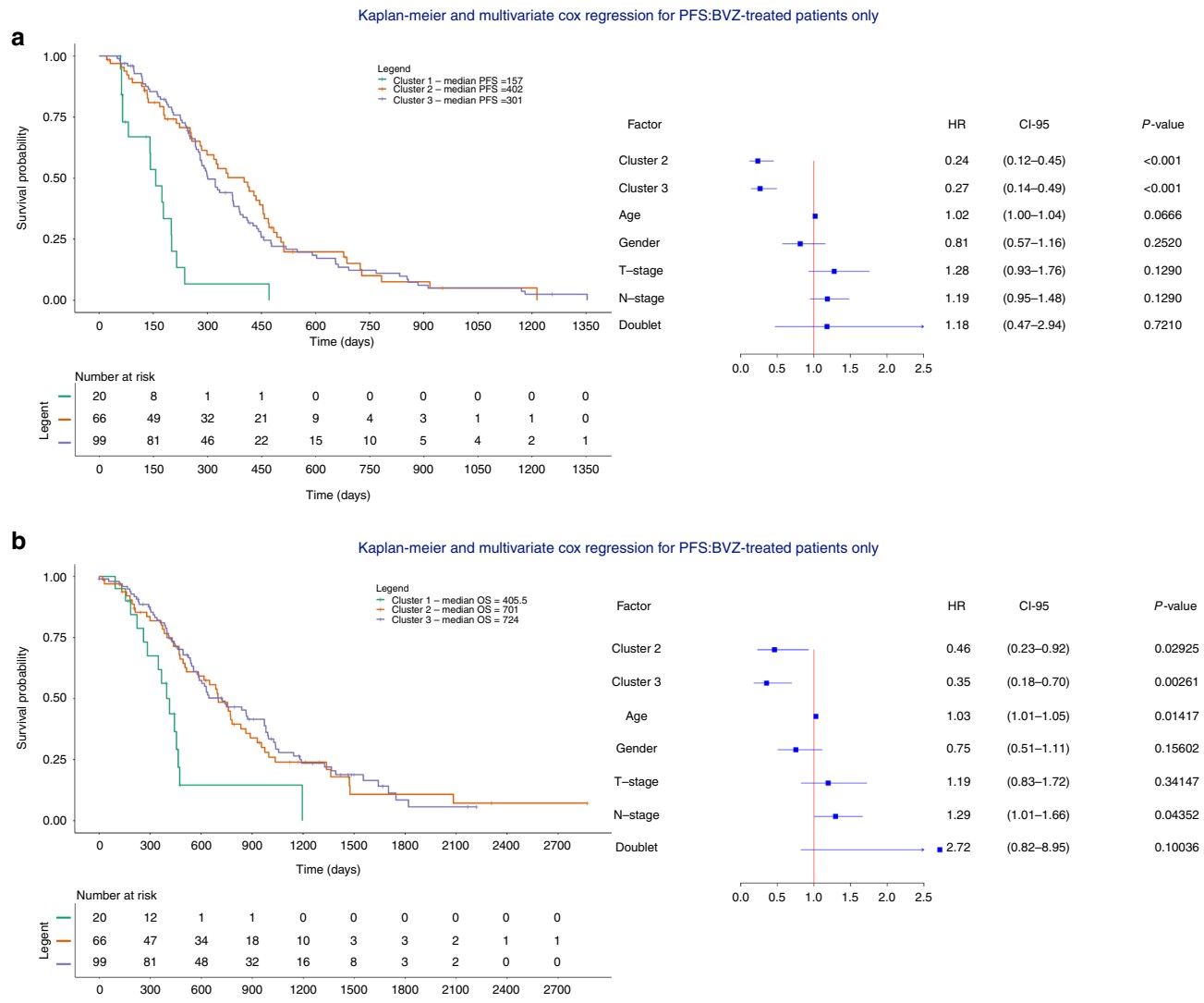

**Fig. 3** Multivariate Cox regression of the different clusters BVZ-treated mCRC samples. **a**, **b** Kaplan-Meier plots and multivariate Cox regression with hazard ratios, 95% confidence intervals and P-values for CNA clusters are shown while correcting for the relevant covariates in mCRC samples treated with chemotherapy + BVZ ($n = 185$). Cluster 1 is considered a reference. Doublet stands for mono-chemotherapy (FP) or a combination of chemotherapy (FP-OX, FP-IRI). Clusters 2 and 3 are correlated with significantly better PFS (**a**) and OS (**b**) independent of clinical covariates. Doublet chemotherapy is not significant in either of the two analyses

respectively 149 and 85 days compared to chemotherapy alone. Similar results were obtained when combining patients from clusters 2 and 3 into one group ($P = 2.49 \times 10^{-3}$, HR = 0.68, CI 0.53–0.87, Cox regression) (Fig. 4d). Power calculations comparing tumors from cluster 1 versus cluster 2 and 3 for chemotherapy ± BVZ revealed 92% power to detect a HR = 0.68. Likewise, BVZ combination therapy showed improved OS for patients with tumors belonging to clusters 2 and 3 (Supplementary Figure 10).

Since MSI tumors form a biologically distinct entity in mCRC, characterized by exceptional therapeutic benefits from checkpoint immunotherapy, we assessed whether cluster 1 tumors were resistant to BVZ because of this enrichment. When stratifying tumors from cluster 1 into MSI ($n = 11$) and non-MSI ($n = 18$) tumors and assessing response of these tumors to BVZ (Fig. 5a, b), we failed however to see a difference between both groups. Also, when correcting the multivariate Cox regression for hypermutation status or excluding hypermutators from the

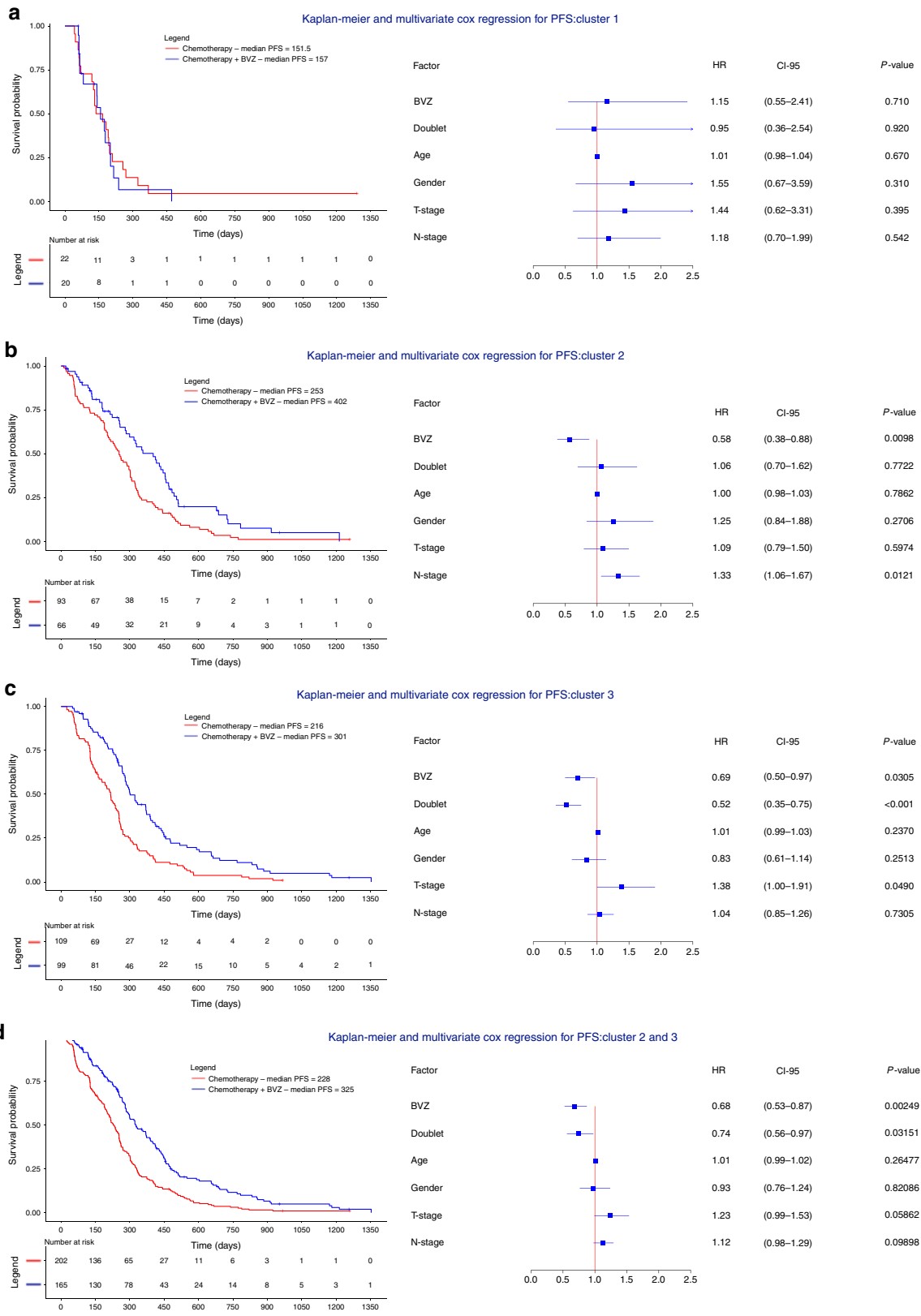

**Fig. 4** Multivariate Cox regression assessing the effect BVZ while stratifying for CNA cluster membership. **a**–**d** Kaplan-Meier plots and multivariate Cox regression with hazard ratios, 95% confidence intervals and P-values are shown while correcting for the relevant covariates in mCRC receiving chemotherapy + BVZ while stratifying for CNA cluster 1 (**a**), cluster 2 (**b**), cluster 3 (**c**) and cluster 2 + 3 versus cluster 1 (**d**). Effects were only significant for the latter 3 comparisons. Doublet stands for mono-chemotherapy (FP) or a combination of chemotherapy (FP-OX, FP-IRI)

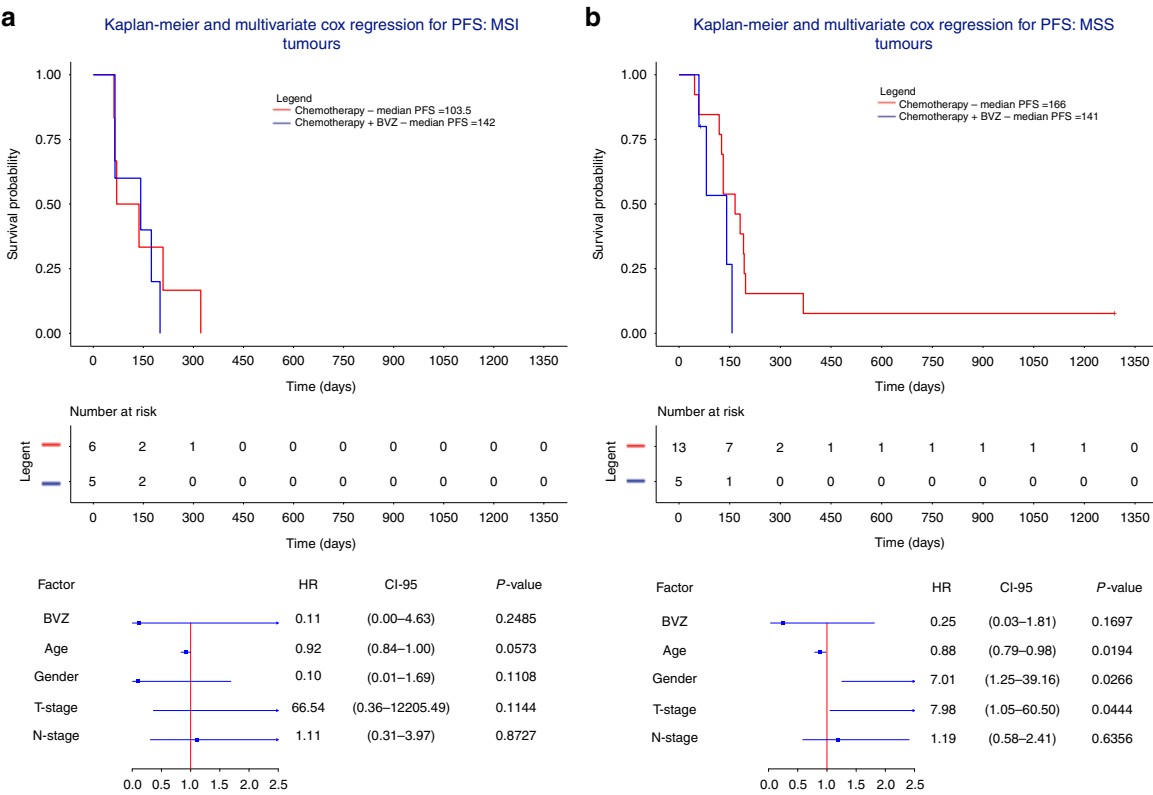

**Fig. 5** Multivariate Cox regression in microsatellite-instable (n = 11) or -stable (n = 18) cluster 1 tumors. **a**, **b** Kaplan-Meier plots and multivariate Cox regression with hazard ratios, 95% confidence intervals and P-values are shown while correcting for the relevant covariates in CNA cluster 1 tumors receiving chemotherapy + BVZ stratified for tumors that were either MSI-positive (n = 11) (**a**) or MSI-stable (n = 18) (**b**). In none of the two groups there was a significant treatment effect of BVZ

analysis, effects were not significantly altered (Supplementary Figure 11). This suggests that copy number stable tumors benefit from BVZ independently of their MSI status.

**High CIN predicts response to BVZ.** Based on our observations that CNA clusters characterized by intermediate-to-high CIN levels exhibited improved response to BVZ, we sought to define the optimal CIN threshold to define which tumors were most likely to respond to BVZ. We stratified patients in two groups based on the proportion of chromosomes affected by CNAs. The optimal threshold was observed at a CIN threshold, where ≥25% of regions were affected by CNAs. Using this threshold, 96% of patients from cluster 1 were defined as CIN-low and 98% of patients from clusters 2 and 3 as CIN-high. When comparing CIN-high versus CIN-low tumors between patients receiving BVZ combination treatment, the former was characterized by significantly improved PFS ($P = 4.31 \times 10^{-4}$; HR = 0.35; CI 0.20–0.63, Cox regression; Fig. 6a). This was not observed when CIN-high versus CIN-low tumors were compared in patients treated with chemotherapy alone (Fig. 6b). Additionally, comparing CIN-high tumors receiving BVZ versus chemotherapy alone revealed a significantly improved survival ($P = 6.38 \times 10^{-3}$; HR = 0.70; CI 0.53–0.90, Cox regression; Fig. 6c). However, no such correlation was observed when assessing CIN-low patients (Fig. 6d). Similar effects were observed for OS (Supplementary Figure 12). An interaction analysis further confirmed CIN as a predictive marker of BVZ treatment outcome (P for interaction = $3.33 \times 10^{-2}$; HR = 0.49; CI 0.26–0.95; Supplementary Figure 13).

**Validation of CIN as a predictive marker of BVZ response.** To replicate these findings, we collected material from 106 mCRC patients participating in the MoMa clinical trial (NCT02271464).

After histopathologic examination of tumor content followed by DNA extraction, we were able to successfully generate CNA profiles on 81 tumors. All patients were treated with BVZ combination therapy (FP, OX and IRI with BVZ) followed by maintenance with BVZ (n = 44) or BVZ plus metronomic chemotherapy consisting of capecitabine and cyclophosphamide (n = 37). Patient characteristics are summarized in Table 1 and Supplementary Table 1.

Next, by using a random forest approach we developed a classifier to assign each of the 81 tumors to one of the 3 CNA clusters (see Methods for additional details and Supplementary Table 3 and Supplementary Data 3). After correcting for relevant covariates, tumors classifying as cluster 2 or 3 showed improved PFS compared to those from cluster 1 ($P = 1.24 \times 10^{-2}$; HR = 0.30; CI 0.12–0.77 and $P = 1.08 \times 10^{-2}$; HR = 0.32; CI 0.14–0.77, Cox regression, respectively for PFS; Fig. 7a). Specifically, patients with tumors belonging to cluster 2 and 3 were characterized by an increase in median PFS of 82 and 75 days respectively. Also, when classifying tumors as CIN-high or CIN-low based on the 25% threshold, we observed a significantly improved PFS for CIN-high tumors treated ($P = 1.99 \times 10^{-3}$; HR = 0.28; CI 0.12–0.63, Cox regression) (Fig. 7b).

**Overlap between CNA and CMS clusters in CRC.** We then correlated our CNA clusters with gene expression data and CRC consensus molecular subtypes (CMS). Since expression data were not available for APD or CAIRO tumors, expression data and CMS subtypes were only assessed for 362 (out of 499) TCGA tumors. Gene set enrichment analysis (GSEA) and MSigDB analysis for 50 hallmark pathways applied to differentially expressed genes between CNA clusters revealed that cluster 1 tumors were characterized by a strong immune-activated

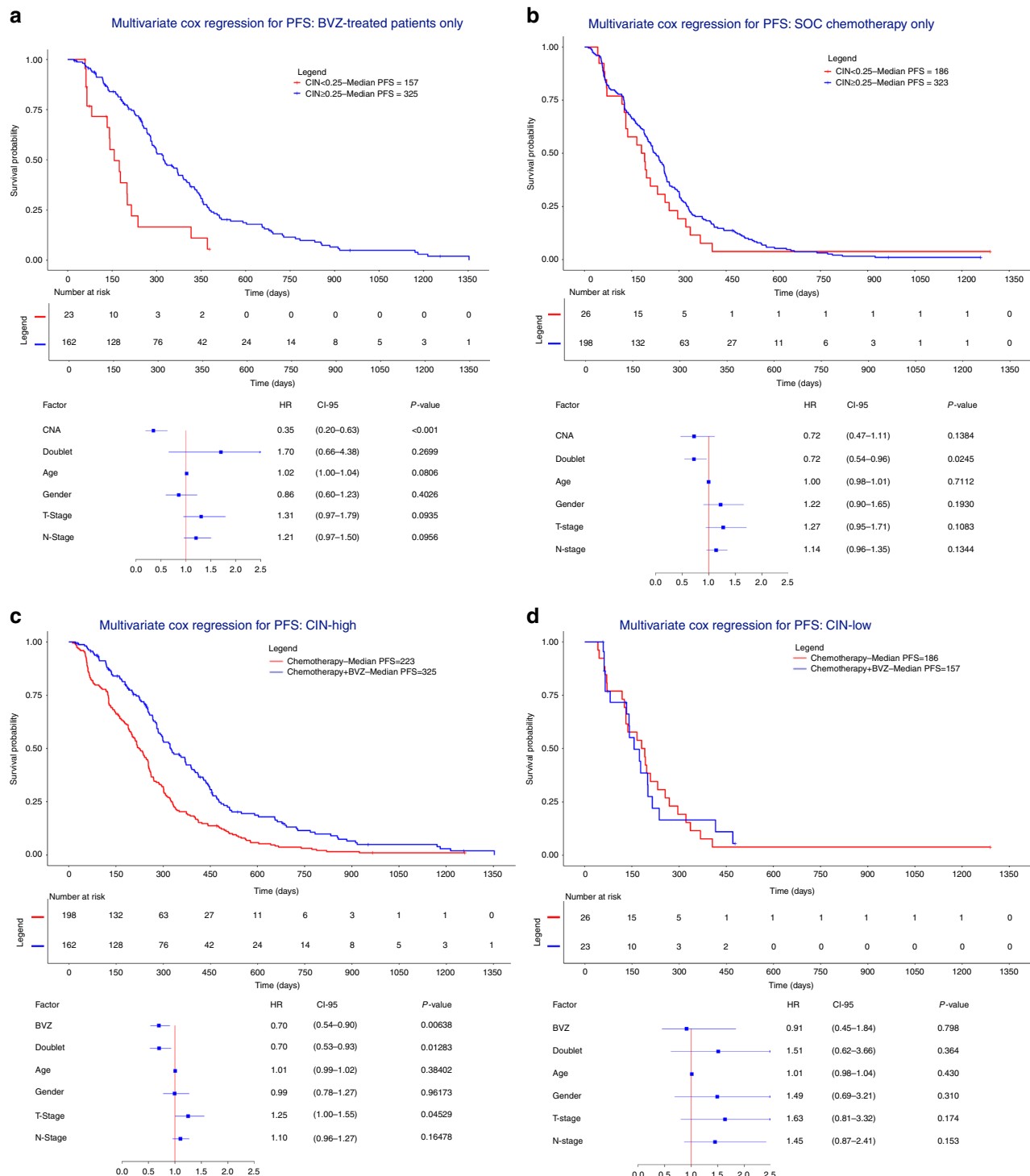

**Fig. 6** Multivariate Cox regression assessing the effect BVZ in CIN-high and CIN-low tumors. **a–d** Patients (*n* = 409) were stratified in CIN-high and CIN-low tumors based on CNAs. CIN-high tumors are defined as having ≥25% of the chromosomal regions affected by CNAs. Kaplan-Meier and multivariate Cox regression with hazard ratios, 95% confidence intervals and *P*-values for PFS are shown while correcting for the relevant covariates for **a** patients treated with BVZ having high CIN versus low CIN, **b** patients treated with standard-of-care chemotherapy having high CIN versus low CIN, **c** patients with high CIN comparing chemotherapy + BVZ versus chemotherapy alone, and **d** patients with low CIN comparing chemotherapy + BVZ versus chemotherapy alone

microenvironment, while cluster 2 and 3 tumors were characterized by angiogenesis, epithelial-to-mesenchymal transition and inflammatory response pathways (Fig. 7c). When assessing CMS signatures, 82.5% tumors from cluster 1 were CMS1 (55.7%) or CMS3 (26.8%), while 77.4 and 91.4% of cluster 2 or 3 tumors respectively, were CMS2 (52.6 and 53.1%) or CMS4 (24.8

and 38.3%) tumors (Fig. 7d). CMS1 tumors indeed display low CIN levels, are often hypermutated or MSI and enriched for *BRAF* mutations, while CMS3 tumors have a mixed MSI status and low abundancy of CNAs. CMS2 and CMS4, on the other hand, have a high level of CIN and contain few hypermutated tumors.

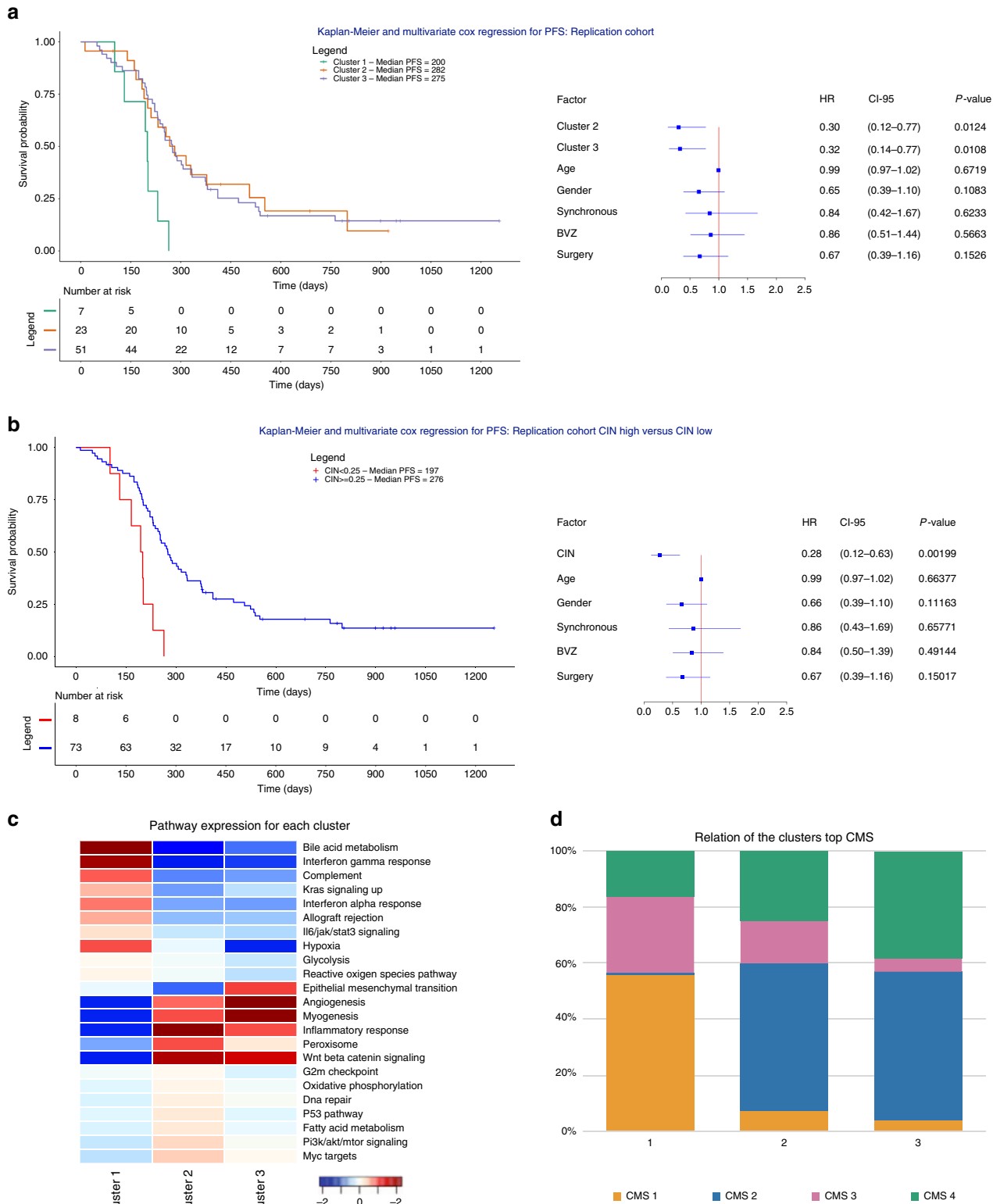

**Fig. 7** Replication cohort, pathway expression and overlap with the consensus molecular subtypes. **a** Application of the random forest classification model to the replication cohort ($n = 81$) classified the samples in 3 different CNA clusters. Multivariate Cox regression with hazard ratios, 95% confidence intervals and *P*-values are shown for the 3 CNA clusters while correcting for the relevant covariates. Both CNA clusters 2 and 3 were characterized by improved PFS. **b** Multivariate Cox regression with hazard ratios, 95% confidence intervals and *P*-values are shown for the high CIN versus low CIN tumors while correcting for the relevant covariates. High CIN tumors were characterized by improved PFS. **c** Heatmap plot showing which pathways were overrepresented in genes differentially expressed in one cluster versus all other clusters. **d** Overlap between CNA clusters and the CRC molecular subtypes. CMS subtypes could only be established for 362 (out of 499) TCGA tumors for which expression data were available. 82.5% tumors from cluster 1 were CMS1 (55.7%) or CMS3 (26.8%), while 77.4 and 91.4% of cluster 2 or 3 tumors respectively, were CMS2 (52.6 and 53.1%) or CMS4 (24.8 and 38.3%)

**Validation of CNA/CMS subtypes as a biomarker in xenografts**. To provide additional independent confirmation of our findings and further explore the impact of CNA clusters and CMS subtypes on BVZ response, a panel of seven xenografts representing each CMS subtype was treated with FOLFOX and B20 (mouse avastin) for 4 weeks[23,28,29]. The following cell lines were employed: Lovo (CMS1, MSI), HT29 (CMS3, MSS), HROC24 (CMS3, MSI), Colo205 (CMS2, MSS), SW620 (CMS2, MSS), DiFi (CMS2, MSS), and SW480 (CMS4, MSS). Due to significant tumor ulceration and necessary early euthanization of animals, the HT29 xenograft was excluded from further analysis. CNA profiling was performed as described for the MoMa samples (HT29 and COLO205: cluster 3; SW620 and SW480: cluster 2 and HROC24, LOVO and DIFI: cluster 1). Each cell line was implanted subcutaneously and treated, as described (see Methods).

In all CMS2 and CMS4 xenografts, the combination of FOLFOX followed by B20 was significantly ($P < 0.05$) better than FOLFOX alone (Fig. 8, Supplementary Figure 14). Further analysis of tumor size allowed classification of treatment response data based on modified RECIST (mRECIST) criteria[30]. SW480, 620, DIFI, and Colo205 showed significant ($P = 0.0046$, 0.02, 0.0001, and <0.0001, respectively, Student's $t$-test) delay in progression in the combination arm compared to FOLFOX alone, whereas LOVO and HROC24 displayed no significant difference ($P = 0.27$ and 0.54, respectively, Student's $t$-test; Supplementary Figure 14). Furthermore, immunohistochemical staining revealed a significantly reduced microvessel (CD31 and vWF) density for CMS2 and CMS4 subtypes treated with B20 or FOLFOX + B20 compared to CMS1 and 3. No significant differences were noted for treatment with FOLFOX alone (Fig. 8, Supplementary Figure 15). Moreover there was a significant reduction in proliferation (via Ki67) of CMS1, 2 and 4 subtyped cells lines treated with FOLFOX + B20 over vehicle, while CMS3 showed no significant effect after treatment with FOLFOX + B20 (Fig. 8). Similar observations, were made when stratifying cell lines according to CNA cluster membership (with cell lines belonging to CNA cluster 1 failing to respond, while those belonging to CNA clusters 2 and 3 revealing a significant response). Of notice, the CMS4 subtype DIFI cell line had a tetraploid karyotype and was therefore also classified as copy number instable[31].

## Discussion

A high degree of CIN represents a form of genomic instability that is present across most solid tumors and has been associated with poor patient outcome in several cancer types[32,33]. Overall, CIN results from defects in mitosis and pre-mitotic replication stress[34], with mutations in *TP53* and other genes having a permissive role. Somatic CNAs that give rise to CIN are evident in 85% of invasive CRCs, where their stepwise accumulation is known to stimulate tumor initiation and progression[35], either by activating oncogenes or inactivating tumor suppressors genes[36]. Efforts to understand the implications of CNAs in CRC have led to important discoveries with respect to disease prognosis[36], but their role in predicting response to therapy remains largely unexplored. We therefore specifically assessed the relevance of CNAs in predicting outcome of BVZ combination therapy in mCRC.

First, we used CNA profiles from 908 CRC patients to classify tumors in 3 distinct CNA clusters using unsupervised hierarchical clustering. Characterization of these clusters revealed that cluster 1 is enriched for tumors carrying an excess of somatic mutations, including amongst others MSI tumors and tumors with *POLE* and *POLD1* mutations, and is characterized by a low number of

CNAs. The number of tumors classified to cluster 1 was substantially lower for mCRC tumors (10.2%) compared to all 908 CRC tumors involving different stages (18.7%)—a direct result of the lower proportion of MSI tumors that acquire a metastatic phenotype. Indeed, only 3–5% of mCRC tumors are expected to be MSI[37–39]. This provides further proof that cluster 1 does not solely consist of MSI tumors but is also characterized by non-MSI copy number stable tumors.

Next, we observed that copy number instable mCRC tumors demonstrated improved survival compared to copy number stable tumors. When focusing on mCRC patients receiving BVZ, we noticed a markedly decreased HR at the level of PFS and OS in patients from clusters 2 and 3 relative to patients receiving chemotherapy alone. Likewise, when stratifying patients according to cluster, tumors belonging to cluster 1 did not benefit while for tumors belonging to clusters 2 and 3 a significant increase in survival was observed when receiving BVZ combination therapy. Similar effects were observed when considering a CIN threshold of 25%, both in our discovery and replication cohort. Importantly, determination of CNA cluster membership or CIN using low-coverage whole-genome sequencing can be achieved quickly, reliably and cost-effectively using, even when degraded tissue. Indeed, with the advent of non-invasive prenatal diagnosis this technique is routinely used in the diagnostic setting.

CMS is currently the most robust classifier for CRC based on gene expression profiling. Additionally, there is accumulating evidence that these subtypes may predict clinical outcome[40,41]. Our analysis revealed an overlap between CMS subtypes and CNA clusters, with an enrichment of CMS1/3 in CNA cluster 1 and CMS2/4 in CNA clusters 2 and 3. The mesenchymal subgroup, known as CMS4, is characterized by tumors with a high CNA load and more pronounced VEGF and VEGFR activation levels, hence displaying a pro-angiogenic and pro-inflammatory phenotype[23,42]. This was confirmed in pathway analyses of differentially expressed genes in cluster 2 or 3. It is therefore conceivable that the anti-angiogenic BVZ is more effective in these subtypes. To indeed confirm the predictive effect of CNA and CMS subtyping on BVZ outcome, we employed a panel of CRC cell lines for which CMS subtype was previously determined[28] and for which we a priori assessed CNA cluster membership. We observed a significantly enhanced response to the anti-angiogenic B20 or FOLFOX plus B20 compared to FOLFOX alone in grafted CMS4 and CMS2 cell lines. Notably, all these cell lines were also classified as CIN-high tumors. These experimental xenograft data thus confirm our findings in BVZ-treated patients. Remarkably, Lenz et al. recently revealed that CMS1 colon cancer benefits more from BVZ-based treatment than cetuximab-based treatment[24]. Although the latter study compares BVZ effects to different control groups than in this study (wild-type RAS patients treated with cetuximab versus standard-of-care chemotherapy-treated patients in this study), these findings highlight the need for additional studies to confirm our findings.

Recently Le et al. showed how MSI mCRC tumors, which typically are associated with high tumor mutational burden, respond extremely well to PD-1 blockade with pembrolizumab, ultimately leading to the pan-cancer approval of anti-PD-1 therapy for MSI tumors[17]. Nowadays, MSI tumors will therefore first receive anti-PD-1 immunotherapy, rather than BVZ combined with chemotherapy. Our data, which show that tumors characterized by low copy number burden do not benefit from BVZ, thus seem to confirm that anti-PD-1 therapy is a better treatment option for these patients. Furthermore, our data suggest that other CNA cluster 1 tumors that are not MSI, also do not benefit from BVZ therapy and might therefore also be treated with anti-PD-1 therapy[17]. Although this needs to be confirmed in follow-up prospective clinical studies, the use of copy number

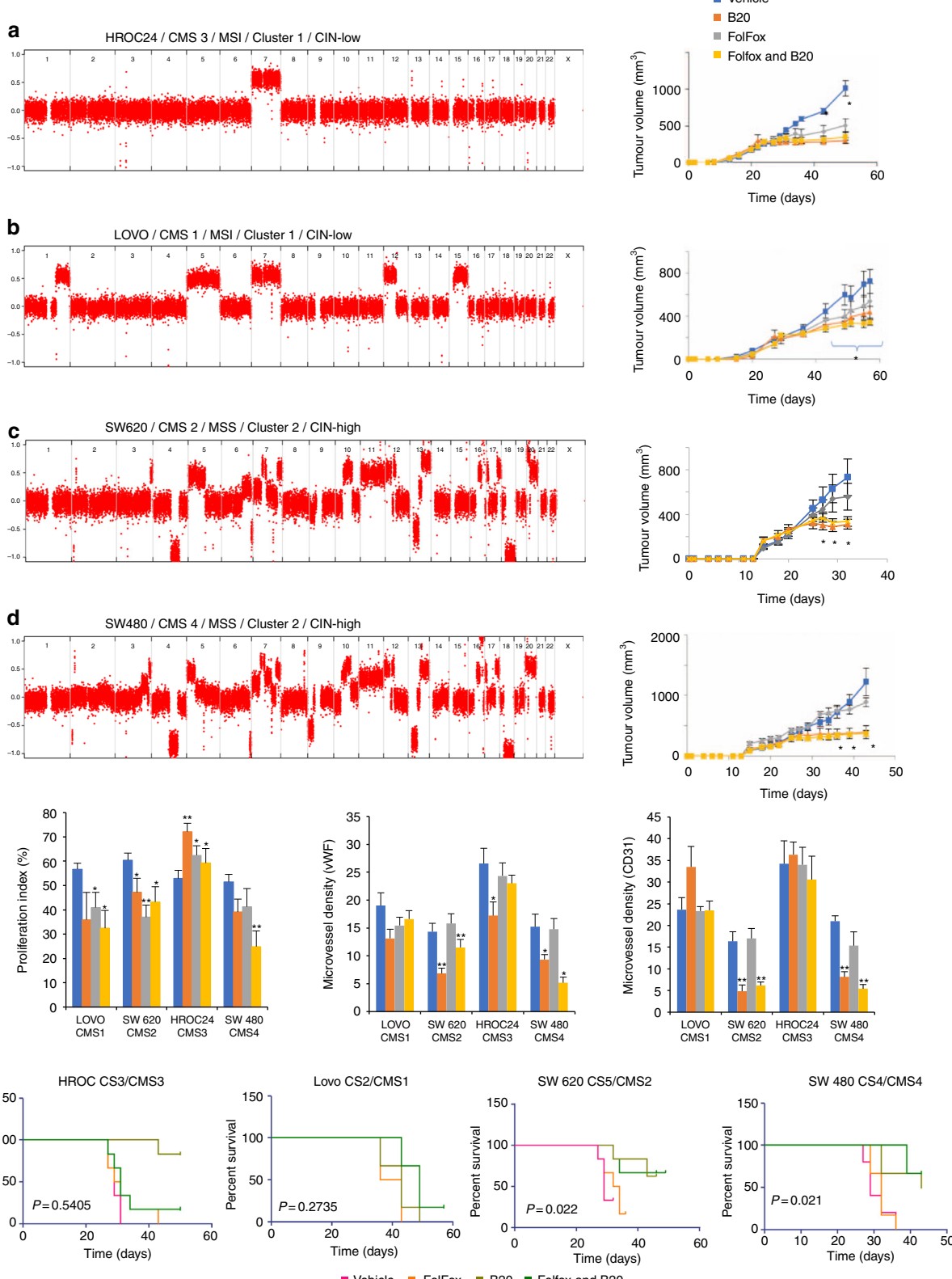

**Fig. 8** In vivo experiments and IHC analyses on xenografts. **a–d** Whole-genome copy number profile and growth curves of xenografts and analysis of the tumor sizes for four out of the seven cell lines. Error bars represent s.e.m. of six animals per group. Student's *t*-test *$p < 0.05$. **e** Immunohistochemical staining to determine the proliferation index (Ki67) and microvessel densities (vWF and CD31). Error bars represent s.e.m. Student's *t*-test *$p < 0.05$, **$p < 0.01$. **f** KM-plots for PFS based on modified RECIST criteria. Regression was defined as a 50% decrease in tumor size and progression as a 35% increase in tumor size

load as an additional biomarker to tumor mutational burden, might become clinically useful. Additionally, our data provide additional insights into the recent observation that chromosome 18q11.2–18q21.1 loss predicts response to BVZ in mCRC[26]. Indeed, our findings suggest that genome-wide instability, rather than the specific loss of one chromosomal region, underlies the association of CNAs with response to BVZ.

One limitation of this study is the lack of available information with respect to tumor sidedness. Right-sided stage III-IV tumors are associated with inferior prognosis, and based on the clinical and biologic characteristics of right-sided tumors, we know that these are more likely to be MSI, carry *BRAF* mutations and represent hypermutators[23,43]. It is likely that CNA cluster 1 might be enriched for right-sided tumors, whereas CNA clusters 2 and 3 could vice versa be enriched for left-sided tumors. Future studies are warranted to confirm these hypotheses. Other limitations are the retrospective nature and potential selection bias in our sample population.

In conclusion, by considering genome-wide CNAs in CRC and by applying subsequent unsupervised clustering, we were able to classify mCRC tumors into CNA subtypes and to relate their response to outcome after BVZ combined with chemotherapy. Tumors that are classified in CNA clusters 2 and 3, and therefore likely correspond to CMS2 or CMS4 subtype tumors, show additional benefit from BVZ treatment when compared to patients from the same cluster receiving chemotherapy only. Hypermutator phenotypes, such as tumors with *POLE* or *POLD1* mutations or micro-satellite instable tumors show no additional benefit from BVZ treatment and importantly also MSS tumors with a stable copy number profile show no additional benefit from BVZ treatment. We therefore propose that high copy number load represents a potential novel biomarker for BVZ response.

## Methods

**Sample collection**. Patients with advanced (locally irresectable or metastatic) CRC commencing combination chemotherapy involving BVZ between July 2004 and April 2012 were included in this study. Particularly, criteria for inclusion were: (1) histologically proven diagnosis of colon or rectum adenocarcinoma, either metastasized or locally advanced and irresectable, and (2) combination chemotherapy with a regimen including bevacizumab at any line of chemotherapy. Tumor tissue from 274 mCRC patients fulfilling these criteria were retrospectively collected from the tissue biobanks of the Royal College of Surgeons in Ireland (RCSI) Beaumont Hospital ($n = 29$), the University of Heidelberg (UHEI), Germany ($n = 107$) and the VU University Medical Center (VUMC) in The Netherlands ($n = 34$)[44]. The follow-up period for the UHEI, VUMC and RSCI cohorts started on July 28, 2004, September 7, 2004, and August 18, 2004, respectively. They ended on December 15, 2014, July 03, 2013, and June 02, 2015, respectively. Follow-up included CT scans or abdominal ultrasound and chest X-ray every 3 months. T-classifications and N-classifications, grading, and localization of the tumor samples were collected by reviewing patients' records and were routinely assessed by different pathologists from the participating centers. Most tumor tissues selected were collected at diagnosis of a metastatic CRC ($n = 166$), but a minority was collected before metastatic disease relapse, i.e., at the time of resection of an early stage CRC (for which they developed a metastatic relapse; $n = 108$). For the latter patients, we considered tumor characteristics collected at the time of resection, while treatment data (involving BVZ plus chemotherapy) and outcome were considered for metastatic disease relapse.

Additionally, DNA was extracted from 108 mCRC tumors collected within the CAIRO2 trial, treating mCRC patients with chemotherapy, as described[45]. Another (replication) cohort ($n = 106$) of mCRC tumors treated with BVZ and chemotherapy was collected within the MOMA clinical trial (NCT02271464) and provided to us by the University of Pisa, Italy.

Informed consent was obtained from each patient and institutional review board approval was obtained from the responsible ethics committees for all participating study centers. After tissue collection, samples were reviewed by qualified pathologists to reconfirm diagnosis and delineate adjacent normal tissue. Only tumor blocks with (1) at least 30% tumor cell content, as judged by board certified pathologists on a routine hematoxylin and eosin (H&E) staining, (2) sufficient tissue volume in order to allow successful DNA isolation and (3) clinical data (including gender, age, grade, stage and treatment follow-up) were considered. PFS and OS were considered as clinical end points. PFS was defined as the time from start of bevacizumab therapy to progressive disease or death from any cause,

whichever occurred first. Patients stopping bevacizumab therapy due to reasons other than progression or death were censored as of the date of treatment cessation. OS was defined as the time from start of bevacizumab to death from any cause. All patient data were administratively censored after 60 months.

Additionally, for some analyses, we also used publicly available copy number data for a cohort of 205 patients from the CAIRO trial that were treated with Irinotecan-Capecitabine (CAPIRI) or capecitabine (CAP) only (Agilent oligonucleotide hybridization arrays; GSE36864)[13] and a cohort of 499 patients from the TCGA network (http://gdac.broadinstitute.org/).

**DNA isolation**. After pathological examination, 3–10 tissue sections (5–10 μm thickness) collected from Formalin-fixed, Paraffin-embedded (FFPE) tumors were used for DNA extraction. Regions with high tumor content, as well as regions containing only normal cells (as indicated by the pathologist), were macro-dissected from individual slides. Subsequently FFPE tissue sections were depar-affinized using a series of xylene and ethanol washes. The sections were purified and homogenized (by gentle shaking at 400 rpm while incubating in buffer ATL and Proteinase K at 56 °C) to remove fixatives and assist lysis. After depar-affinization and tissue digestion, DNA was further extracted using the QIAamp DNA FFPE Tissue kit (QIAgen) following the manufacturer's instructions. The resulting DNA was quantified using the Picogreen Assay (Life Technologies) to determine the concentration of double-strand DNA. Only samples with a yield of more than 0.5 μg of dsDNA and a concentration >7.5 ng μl$^{-1}$ were selected for further library preparation.

**Low-coverage whole genome sequencing**. Shot-gun whole genome libraries were prepared using the KAPA library preparation kit (KAPA Biosystems). Whole genome DNA libraries from matched normal and tumor tissue samples were created, according to manufacturer's instructions. Before end-repair, a 4-hour incubation step at 65 °C was added to remove as many reversible crosslinks as possible, after which excessive single stranded DNA was removed using Mung-Bean nuclease. The concentration of double-stranded DNA was reassessed using Pico-green and the concentration of adapters used in the ligation step of the library construction was modified based on the DNA measured. For the library enrichment, 5–15 cycles of PCR with intermediate assessment steps were used instead to ensure low adapter dimer content and high library yield.

Following quantification with qPCR, the resulting libraries were sequenced on a HiSeq2500 (Illumina) at low coverage (±0.1×) for shallow-seq. Raw sequencing reads were mapped to the human reference genome (NCBI37/hg19) using Burrows-Wheeler Aligner (BWA v0.5.8a)[46]. Picard (v1.43) was used to remove PCR duplicates. Somatic copy number aberrations (CNAs) were identified by binning the reads in 100 Kb windows, correcting for genomic waves using the PennCNV software package[47] and the resulting number of reads per 100 Kb window were transformed into log R-values. Only samples with more than 1 million mapped reads and a mean absolute pair-wise deviation lower than 0.4 were used in further analyses (Supplementary Note 1). The ASCAT algorithm version 2.0.1[48] was used to segment the raw data and estimate tumor percentages and overall ploidy. Subsequently, GISTIC v2.0[27] was used to identify the most frequent and overrepresented chromosomal aberrations in tumors. A region was considered deleted if the logR value was <−0.1 and amplified when the logR was >0.1. A cutoff q-value of 0.25 was used to select significantly overrepresented SCNAs. SCNAs spanning >70% of a chromosomal arm were defined as whole-arm SCNAs, while SCNAs spanning <70% of a chromosomal arm were considered focal SCNAs. Significant amplified or deleted regions were assigned as homozygous deletion, loss, diploid, gain, or amplification for each sample based on LogR signal and GISTIC output threshold values ($t < −1.3$; $−1.3 \leq t < −0.1$; $−0.1 \leq t \leq 0.1$; $0.1 < t \leq 0.9$; $t > 0.9$ respectively).

**Whole-exome sequencing**. After confirmation of successful library construction, whole-exome enrichment was performed using the SeqCapV3 exome enrichment kit (Roche) following the manufacturer's instructions. The resulting whole-exome libraries were then sequenced on a HiSeq2500 using a V3 flowcell generating 2 × 100 bp paired end reads. Raw sequencing reads were mapped to the human reference genome (NCBI37/hg19) using Burrows-Wheeler Aligner (BWA v0.5.8a)[46] and aligned reads were processed and sorted with SAMtools (v0.1.19)[49]. Duplicate reads were removed using Picard tools. Base recalibration, local realignment around insertions and deletions and single nucleotide variant calling were performed using the GenomeAnalysisToolKit (GATK)[50]. Insertions and deletions were called using Dindel[51]. By subtracting variants and indels detected in the matched germline DNA from those found in the tumor DNA, somatic mutations were selected. Low quality mutations were removed based on mapping quality and coverage. ANNOVAR[52] was used to annotate the remaining mutations and exonic non-synonymous mutations and frame-shift insertions or deletions were selected. Common variants (MAF > 1%) were filtered out using the following databases as described previously[53]: (1) dbSNP version 132, (2) 1000 Genomes Project, (3) Axiom Genotype Data Set, (4) Complete Genomics diversity panel (46 hapmap individuals).

**Statistical analysis**. CNA and mutation calling, and assignment of each tumor to a CNA cluster was done blinded for all treatment data. Consensus clustering was

done using unsupervised Hierarchical Ward clustering on all CRC samples (including those from AngioPredict, CAIRO and TCGA) as well as on all mCRC samples using the packages "ConsensusClusterPlus" and "hclust" in R. As an input, we used recurrent CNAs identified from the GISTIC analysis by applying a sub-sampling size of 80% with 500 repetitions. In the discovery cohort, multivariate survival analysis between and within the different clusters was performed using a Cox regression analysis considering TNM staging and age as numerical factors while gender, cluster membership or CIN as well as chemotherapy backbone were considered categorical variables. A similar analysis was performed comparing CIN-high with CIN-low patients. In the replication cohort, cluster membership or CIN, absence or presence of synchronous metastases, maintenance therapy with BVZ and surgery were used as categorical factors.

For cluster characterization *TP53*, *APC*, *POLE*, *POLD1* and *PIK3CA* mutation status was based on the presence of damaging mutations based on exome sequencing data or data available from TCGA. A hyper-mutator was defined as having more than 10 mutations per $10^6$ bases. Tumors were considered MSI if they had either an immunohistochemical loss for known MMR proteins or damaging mutations in known MSI genes based on exome-sequencing data or data available from TCGA. Similarly, for *KRAS* and *BRAF* we combined staining with damaging mutations detected. Fishers exact test (two-sided *P*-values; $n = 8$) was used to test whether clusters were significantly enriched for certain mutations, MSI-status or a hypermutator phenotype. To test which clinical variables were enriched in particular clusters a ChiSquare test was used (two-sided *P*-values; $n = 5$). Random forest classification was performed using the R-package "randomForest". We performed a 10-fold cross-validation on the original dataset to determine the accuracy of the model. Hereto, we divided the 472 mCRC samples used for the original clustering ten times at random, each time in a training set (90% of the samples) and validation set (10% of the samples) in such a manner that each sample is presented only once in the whole of ten validation sets. Next a random forest classifier was generated using the training data. We then applied this classifier to the validation data to determine the models' accuracy. For each tree, the prediction error rate on the out-of-bag portion of the data is recorded. Then the same is done after permuting each predictor variable. The difference between the two are then averaged over all trees and normalized by the standard deviation of the differences.

For all in vivo experiments, animal numbers were calculated using a power of 80% ($\beta = 0.8$) and an alpha ($\alpha$) of 0.05 and was approved by both local and national animal ethical committees. Statistical analyses of tumor growth curves were performed by two tailed Student's *t*-test with 5 degrees of freedom for each cell line unless otherwise stated. All Kaplan-Meier curves for in vivo tumor progression were statistically analyzed by the log rank test. All reported *P*-values are two-sided unless otherwise stated.

**Cell culture**. Previously subtyped human colorectal cell lines (Lovo RRID: CVCL_0399, HT29 RRID: CVCL_0320, SW480 RRID: CVCL_0546, SW620 RRID: CVCL_0547, Colo205 RRID: CVCL_0218, (all from ATCC Manassas, Virginia USA), DiFi, RRID: CVCL_6895 (donated by Dr. Robert J. Coffey, Jr., M.D of the Vanderbilt University Medical Centre, Nashville, Tennessee USA) and HROC24 RRID: CVCL_1U80 (Cell Line Services Eppelheim, Germany)) were cultured in DMEM/F12 (Sigma), supplemented with 10% (v/v) foetal bovine serum (FBS, Sigma), 100U mL$^{-1}$ penicillin and 100 µg mL$^{-1}$ streptomycin (Sigma) and 2mM L-glutamine (Sigma), in 5% CO2/95% air at 37 °C. Cells were passaged at least three times and tested for the presence of mycoplasma and mouse pathogens (IMPACT II IDEXX, Hoofddorp, The Netherlands) before implantation into mice.

**Animals**. Female Balb/C $^{nu/nu}$ mice ($n = 24$ per cell line, 6–8 weeks, Charles River Laboratories, Sandwich, UK) were housed in groups of 3–5, maintained on a 12 h light/dark cycle, with free access to standard rodent chow and water. Animal experiments conformed to guidelines from Directive 2010/63/EU of the European Parliament on the protection of animals used for scientific purposes. Experiments were licensed and approved by the Health Products Regulatory Authority Ireland (HPRA) project authorisation number AE18982/P100. Protocols were also reviewed by University College Dublin Animal Research Ethics Committee. A one week animal acclimatisation period was allowed prior to beginning studies. Balb C$^{nu/nu}$ mice were implanted with previously subtyped colorectal cell lines in the right flank at various concentrations from $5 \times 10^6$ to $1 \times 10^7$ cells. Tumors were allowed to develop until they reached on average 250 mm$^3$. Subsequently, animals were randomly divided into groups ($n = 6$) and treated with either vehicle (5% glucose and PBS) or the previously determined clinically relevant doses of FOLFOX [Folinic acid 13.4 mg kg$^{-1}$, 5-FU: 40 mg kg$^{-1}$, Oxaliplatin: 2.4 mg kg$^{-1}$], IP once a week. 24 h after the FOLFOX+B20 (10 mg kg$^{-1}$) was administered IP once a week either alone or in combination for a maximum of 4 weeks. Tumors were measured twice weekly by callipers by an investigator blinded for the treatment. Any tumor that reached 15 mm or more in any dimension during the study, the animal was euthanized. After 4 weeks all remaining animals were euthanized and their tumors were fixed in 10% formalin for immunohistochemical processing.

**Immunohistochemistry (IHC)**. Four cell lines representing each of CMS subtype 1–4 were selected for IHC analysis with DAB probes for the cell proliferation

marker Ki67 (1:150 Rabbit α-KI67 Merck Cat #AB9260, heat mediated antigen unmasking), the blood vessel marker vonWillebrand Factor (vWF) (1:75 Rabbit α-mouse vWF Abcam #AB6994, heat mediated antigen unmasking) and the blood vessel marker CD31 (1:25 Rabbit α-mouse CD31 Santa Cruz Cat #SC1560, heat mediated antigen unmasking) in 4 cell lines representing each of the CMS subtypes (CMS1: LOVO, CMS2: SW620, CMS3: HROC24, CMS4: SW480). Three xenografts per cell line were analyzed with a minimum of four images per xenograft. Images for Ki67 were analyzed by color deconvolution in Image J and counting all positive brown nuclei and images for vWF and CD31 were analyzed by applying a 15000 pixel$^2$ grid over the image in Image J and counting the number of times positive vessels cross the grid.

## Data availability

The sequencing data are deposited at the EMBL-EBI under accession code EGAS00001002617 and are available under restricted access.

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

## Acknowledgements

The ANGIOPREDICT project was funded by the European Commission Framework Programme Seven (FP7) initiative under contract No. 278981 'ANGIOPREDICT' (www.ANGIOPREDICT.com). A.T.B. is supported by Science Foundation Ireland under grant 13/CDA/2183. S.D. was supported by the Irish Cancer Society Fellowship award CRF13DAS. M.M. and H.P. are supported by the Flemish Research Foundation 'Kom op tegen kanker' (grant 419.052.173). D.L. is supported by the FWO-F (grant G070615N). M.P.E. was supported by grants from the State of Baden-Württemberg for "Center of Geriatric Biology and Oncology (ZOBEL)—Perspektivförderung" and "Biology of Frailty —Sonderlinie Medizin". We are grateful to Genentech for providing B20 antibody for xenograft studies. D.O. and S.D. are supported by Science Foundation Ireland under grant 15/CDA/3438. H.P. is a Senior Clinical investigator of the Belgian Foundation against Cancer.

## Author contributions

D.L. and A.T.B. conceived the study. T.G., J.B., H.P., A.B., V.M., M.P.E., N.C.T.G., C.C., H.P., B.Y., C.H., F.L., B.H., E.K, D.M., H.V.M, N.M., F.L., M.M and O.B. provided the clinical specimens and data and performed histo-pathological assessment of samples. D.S., I.S.M, J.D., B.B., S.D., B.F., R.K., D.M., J.B., H.V.M, N.M., J.P., M.M. and M.P.E. processed the samples. D.S., S.D., R.K. and N.C.T.G. performed the low-coverage whole-genome sequencing experiments. D.S., S.D. and R.K. performed the whole-exome sequencing experiments. D.S., J.D., B.B. and B.M. analyzed the whole-exome and low-coverage whole-genome sequencing data. I.S.M. and J.B performed the in-vivo experiments and IHC. D.S., I.S.M., D.O, J.D., B.B., M.K., C.J.A.P, A.T.B. and D.L. reviewed and assembled all the results. D.S., I.S.M., J.D., B.B., S.D., T.G., B.M., M.P.E., N.C.T.G., B.Y., W.G., J.P., D.O., F.L., A.T.B and D.L. interpreted the results. D.S., I.S.M., D.O., A.T. B. and D.L. wrote the manuscript. All authors reviewed, edited and approved the final draft.

## Additional information

**Competing interests:** D.L., D.S., and A.T.B. are named as inventors on a patent related to this work (WO 2017/182656). The remaining authors declare no competing interests.

Dominiek Smeets[1,2], Ian S. Miller [3], Darran P. O'Connor [4,5], Sudipto Das[4,5], Bruce Moran [5], Bram Boeckx[1,2], Timo Gaiser[6], Johannes Betge [7], Ana Barat[3], Rut Klinger[5], Nicole C.T. van Grieken[8], Chiara Cremolini[9], Hans Prenen [10,11], Massimiliano Mazzone[1,12], Jeroen Depreeuw[1,2,13], Orna Bacon[3], Bozena Fender[14], Joseph Brady[15], Bryan T. Hennessy[16], Deborah A. McNamara[16], Elaine Kay[17], Henk M. Verheul[18], Neerincx Maarten[18], William M. Gallagher[5,14], Verena Murphy[19], Jochen H.M. Prehn[3], Miriam Koopman[20], Cornelis J.A. Punt[21], Fotios Loupakis[22], Matthias P.A. Ebert[7], Bauke Ylstra [8], Diether Lambrechts[1,2] & Annette T. Byrne[3,5]

[1]VIB Center for Cancer Biology, VIB, Herestraat 49, 3000 Leuven, Belgium. [2]Department of Human Genetics, University of Leuven (KULeuven), Herestraat 49, 3000 Leuven, Belgium. [3]Department of Physiology & Medical Physics, Royal College of Surgeons in Ireland, 31A York Street, Dublin D2, Ireland. [4]Department of Molecular and Cellular Therapeutics, Royal College of Surgeons in Ireland, 123 St.Stephen's Green, Dublin D2, Ireland. [5]UCD School of Biomolecular and Biomedical Science, UCD Conway Institute, University College Dublin, Dublin D4, Ireland. [6]Institute of Pathology, University Medical Center Mannheim, University of Heidelberg, Theodor-Kutzer-Ufer 1-3, 68167 Mannheim, Germany. [7]Department of Medicine II, University Hospital Mannheim, Heidelberg University, Theodor-Kutzer-Ufer 1-3, 68167 Mannheim, Germany. [8]Department of Pathology, Cancer Center Amsterdam, Amsterdam UMC, Vrije Universiteit Amsterdam, De Boelelaan 1117, 1081 HV Amsterdam, The Netherlands. [9]Department of Translational Research and New Technologies in Medicine and Surgery, University of Pisa, Istituto Toscano Tumori, Lungarno Antonio Pacinotti, 43, 56126 Pisa, Italy. [10]Department of Oncology, University Hospital Antwerp, Edegem 2650, Belgium. [11]Center for Oncological Research, Antwerp University, 2650 Edegem, Belgium. [12]Department of Oncology, University of Leuven (KULeuven), Herestraat 49, 3000 Leuven, Belgium. [13]Department of Obstetrics and Gynecology, Division of Gynecologic Oncology, University Hospitals Leuven, KU Leuven, Herestraat 49, 3000 Leuven, Belgium. [14]OncoMark Limited, NovaUCD, Belfield Innovation Park, Dublin D4, Ireland. [15]Veterinary Pathobiology, School of Veterinary Medicine, University College Dublin, Stillorgan Rd, Belfield, Dublin D4, Ireland. [16]Department of Surgery, Beaumont Hospital, Beaumont Rd, Beaumont, Dublin D9, Ireland. [17]Department of Pathology, Beaumont Hospital, Beaumont Rd, Beaumont, Dublin D9, Ireland. [18]Department of Medical Oncology, Cancer Center Amsterdam, Amsterdam UMC, Vrije Universiteit Amsterdam, De Boelelaan 1117, 1081 HV Amsterdam, The Netherlands. [19]Cancer Trials Ireland, Innovation House, Old Finglas Road, Dublin D9, Ireland. [20]Department of Medical Oncology, University Medical Center Utrecht, Utrecht University, Heidelberglaan 100, 3584 CX Utrecht, The Netherlands. [21]Department of Medical Oncology, Amsterdam UMC, University of Amsterdam, Meibergdreef 9, 1105 AZ Amsterdam, The Netherlands. [22]Oncologia Medica 1, Istituto Oncologico Veneto, Istituto di Ricovero e Cura a Carattere Scientifico, IRCCS, Via Gattamelata, 64, 35128 Padova, Italy. These authors contributed equally: Dominiek Smeets, Ian S. Miller, Darran P. O'Connor. These authors jointly supervised this work: Diether Lambrechts, Annette T. Byrne.

