## [Peer Review File · Nature Communications]

Reviewers' comments:

Reviewer #1 (Remarks to the Author):

In this manuscript, Sweets and colleagues separate colorectal cancers into three subtypes based on their copy number profiles. Cluster 1 features rare copy number alterations and enrichment of hypermutators, whereas cluster 2 and 3 are characterized by high levels of copy number alterations affecting more than 75% of the genome. This three-cluster classification is largely replicated in metastatic tumors. The major finding of this work is that tumors of clusters 2 and 3 have better responses to bevacizumab, an angiogenesis inhibitor that has been tested on various types of cancer. They validate the association in an independent cohort (MoMa trial), and furthermore, in xenograft models. They also compare their clusters to consensus molecular subtypes of colorectal cancer. The manuscript is clinically oriented, and the message is clearly conveyed. The major criticism from me is that the manuscript lacks mechanistic insights into the clusters, plus a few technical concerns, see below.

Major

1. Three datasets were used to derive the three clusters, their own (APD), CAIRO, and TCGA. However, there is no indication in the manuscript if batch effects and cohort differences have been controlled for. Do cases from different sources cluster together in the final clusters? And do different patient cohorts have similar outcome, age, sex and grade distributions? And if doing GISTIC separately, do they capture the same amplification and deletion regions?
2. Despite being consider metastatic tumors, 106 cases from the APD cohort were collected prior to metastasis. Their copy number data really reflect the genome of the primary tumor but not the metastasis stage. It is unclear how these 106 cases are distributed across the 3 clusters, and furthermore, when bevacizumab was administered to these patients compared with tumors collected after metastasis. It is possible that the better response to bevacizumab for clusters 2 and 3 are due to the fact that they are more advanced thus have increased hypoxia and angiogenesis. At the copy number level, they might accumulate more copy number changes due to their longer progression time. It is thus useful to compare these 106 cases with the rest of the cohort to see if they have a quieter genome, and if they are given bevacizumab at a different time point relative to sample collection (so to explain their worse outcome).
3. In their attempt to replicate the clusters using mCRC cases only, did they re-do GISTIC or did they use the same GISTIC output from the 908 CRCs? If some events are primary tumor specific, it would not make sense to include them in the mCRC analysis.
4. Most of hypermutators are classified into cluster 1, raising the question that whether hypermutation is a relevant factor here. The authors did compare MSI vs MSS tumors in cluster 1 regarding response to bevacizumab, and suggested it was not significant. However, the sample sizes of both groups are very small (MSI, n=11; MSS, n=18). It is likely that the absence of significance is due to the fact that they do not have the statistical power due to the small cohorts. Why not include hypermutation in the multivariate analysis? Or what is the survival differences and response to bevacizumab between cluster 1 and cluster 2&3 if they exclude hypermutators in the comparison?
5. The authors failed to provide any insights into the biological differences of the three clusters other than comparing mutation frequencies of a few genes and several histopathological parameters. Excluding hypermutators (MSI plus POLE/POLD1), is there still an enrichment for PIK3CA and BRAF mutations in cluster 1? Did they compare the expression of the three clusters using TCGA data? Based on the copy number pattern, it appears what differs between cluster 2 and 3 is DNA loss – cluster 3 has many more recurrent losses than cluster 2. What are these losses, and is there any genes/regions consistently lost in cluster 3 but not 2? Can they provide a IGV screenshot in the supplement so that we can appreciate the copy number landscape of the three groups?
6. In the survival analysis of the validation cohort (Figure 7), why is disease stage is not controlled for as a covariate, similar to the discovery cohort?

Minor

7. Cohort sizes are often inconsistent throughout the manuscript. While the authors initially suggested in "study population" that 278 tumors were collected, table 1 indicates the total number is 215. I assume these 215 cases were those who were subjected to low pass sequencing, but in the table the numbers do not add up to 215 (for instance, 79 females, 134 males, total 213). I guess this might be due to missing info. Nevertheless, the authors need to make a note of it.

8. In the section "Clinical and genomic characterization of CNA clusters in mCRC", paragraph 2, the manuscript states that "a multivariable Cox regression to assess prognostic effects of the 3 CNA clusters in mCRC revealed that both CNA cluster 2 and 3 correlated with poor outcome." This is apparently at odds with their data (Figure 2c and 2d), which show the opposite. Even the odds ratio in the next sentence suggests otherwise.

9. In the section "Patients in CNA cluster 2 and 3 benefit from BVZ combination therapy", they write "In non-BVZ treated patients (n=224), hazard ratios were not significant, neither for PFS (HR=0.57, P=2.4x10⁻², CI=0.35-0.93 and HR=0.72, P=0.18, CI=0.45-1.16 for cluster 2 and 3 respectively) nor OS (P=0.14, HR=0.66, CI=0.38-1.16 and P=0.53, HR=0.84, CI=0.49-1.44 for cluster 2 and 3 respectively),...". Why is p=2.4x10⁻² not significant?

10. I would suggest them to merge the section of comparison with consensus subtype to the section of cluster identification, because I think the flow reads better.

Reviewer #2 (Remarks to the Author):

The manuscript describes quite a lot of background information in a larger data set and then focuses on BVZ-treated patients. Most of these data confirm previous findings, which is fine. The BEV sample set, on which the manuscript focuses, actually involves a modest size number of samples (150-200 approximately from Table 1). There is no interaction P value presented for BVZ x CIN in the main analysis and this critical omission makes the major conclusion of the study untenable. (Significant P values in one group but not another are not sufficient here). The interaction P value (p=0.33) presented is only after groups 2 and 3 have been re-assigned using a modified definition of high and low CIN using an "optimum" cut-point. This analysis is an important concern.

Further comments:

The sample set is very heterogeneous in terms of origins, therapies and follow-up.

The lack of much of the clinicopathological data from the BEV set and of the treatment data used in the wider data set is a concern.

The inclusion of cancers treated with curative intent and those with metastatic disease at presentation creates problems in terms of the most appropriate measure of outcome to use in the full data set.

The minimum 30% tumour cell content required is below the lower limit used in most studies and could easily lead to lack of sensitivity.

Sequencing quality metrics are lacking.

Some of the figure legends seem incomplete (e.g. what is SOC?)

It is arguable that the non-BEV arm shows some evidence of an association between good survival and CIN in group 2.

No estimation of study power for predicting response to BVZ is provided.

Are there other studies on BVZ that have addressed similar questions in colorectal or other cancers? If so, how do they compare? There is no discussion of these.

Reviewer #3 (Remarks to the Author):

This article is well written and well designed. It provides some very valuable information on the role of CNAs (Copy Number Alterations) as a potential predictive value for the use of bevacizumab (BVZ) in metastatic colorectal cancer. The authors underline that the benefit of adding BVZ to standard of care combination chemotherapy could be restricted to patients, whose tumors present with higher CNAs. They are able to show that higher quantification of CIN is also related with a more extensive benefit of BVZ treatment. A valuable relation to Consensus Molecular Subtypes (CMS) is also given as well as further verification of the differential response according to CNAs to BVZ in xenograft models.

The main limitation of this paper could be the retrospective nature of patient selection for the molecular analysis. To reinforce the validity of their analysis the authors should present this process according to the REMARK criteria. See more details below. The authors should also make some interpretation on the potential consequences of their findings for future research in the field as well as for clinical practice.

See below some recommendations on the specific parts of this manuscript.

The abstract is very clear and delivers a very clean message. Its reading allows a full understanding of the content of the article.

In the introduction the authors clearly express the important of the issue they are willing to analyze as well as the main purpose of their research presented in this paper. Reference 1 is outdated (2014) and it should be substituted for a more recent one.

Results

The study population is well described. The method of sample selection is well described. A reference to the REMARK criteria here would reinforce the good methodology used in this article. See McShane LM, Altman DG, Sauerbrei W, Taube SE, Gion M, Clark GM; Statistics Subcommittee of the NCI-EORTC Working Group on Cancer Diagnostics. Reporting Recommendations for Tumor Marker Prognostic Studies. *Br J Cancer* 2005; 93:387-391.

The section on unsupervised clustering of CNAs does not require an amendment as well as the one on Clinical and Genomic characterization of CNA clusters. However, in the first paragraph of page 11 when they refer to the benefit of bevacizumab in CNA cluster 2 and 3, I wonder if differences in responses (response rate should be given as a proportion or percentage of responding patients) should be better expressed as Odds ratios rather than with Hazard ratios. Check for the same in the second paragraph of page 11 when describing response differences in BVZ treated patients.

The sections on the predictive value of CIN for BVZ treatment and its validation in the MoMa trial cohort do not require any modifications. It is likewise for the last two sections on the overlap with CMS clusters and on the use of xenografts.

Tables and figures are informative and well designed. The supplementary material is also of value.

Discussion

It is perhaps too long and it should be shortened. The authors should focus on their findings as well as its potential usefulness and limitations. In that sense, the first paragraph in page 17 could be maintained. However, in the second one, the example of allelic imbalance of chromosome 18 and the one on the value of MET should be removed because it doesn't add any relevance to this publication. I would recommend jumping from "A continuous effort to understand...to disease prognosis³²" to "We therefore explore the prognostic relevance of CNAs...for BVZ have not yet emerged."

A reference to prenatal diagnosis in the second paragraph of page 19 is not really needed and it should be removed.

In page 20 the references to ICON7 and NSABP C-08 are very speculative and not relevant for the discussion of this paper and therefore should be removed. The reference to van Dijk et al on chromosome 19q loss unless already published should also be removed.

In page 21 and 22 the full paragraph dedicated to discuss the potential value of immunotherapy in CNA cluster 1 is somehow out of the main scope of this article and it has to be significantly shortened. I would keep the potential value of associating check-point inhibitors and anti-angiogenics in cluster 1 patients.

I would agree with the authors that one limitation of this study is the lack of information on tumor sidedness. Not an issue. However, I think it is very relevant to discuss on the potential clinical application of the information provided. The retrospective nature plus the potential selection bias in the sample population of this analysis is also limiting. The article states that CIN represents a novel marker for bevacizumab response, but in what sense can it be applied. Do the authors recommend not to use chemotherapy plus bevacizumab in patients classified as CAN cluster 1? What kind of additional information is needed to make a strong statement on this? A comment on these points should be added to the discussion.

Response letter

Reviewer #1 (Remarks to the Author):

In this manuscript, Smeets and colleagues separate colorectal cancers into three subtypes based on their copy number profiles. Cluster 1 features rare copy number alterations and enrichment of hypermutators, whereas cluster 2 and 3 are characterized by high levels of copy number alterations affecting more than 75% of the genome. This three-cluster classification is largely replicated in metastatic tumors. The major finding of this work is that tumors of clusters 2 and 3 have better responses to bevacizumab, an angiogenesis inhibitor that has been tested on various types of cancer. They validate the association in an independent cohort (MoMa trial), and furthermore, in xenograft models. They also compare their clusters to consensus molecular subtypes of colorectal cancer. The manuscript is clinically oriented, and the message is clearly conveyed. The major criticism from me is that the manuscript lacks mechanistic insights into the clusters, plus a few technical concerns, see below.

Major criticism:

1. Three datasets were used to derive the three clusters, their own (APD), CAIRO, and TCGA. However, there is no indication in the manuscript if batch effects and cohort differences have been controlled for. Do cases from different sources cluster together in the final clusters? And do different patient cohorts have similar outcome, age, sex and grade distributions? And if doing GISTIC separately, do they capture the same amplification and deletion regions?

- We thank the reviewer for raising this valid question. We did assess the 3 clusters in the separate cohorts prior to submitting the original manuscript, but we indeed failed to report the outcome of these analyses. Briefly, we observed that the frequency of the 3 clusters was more or less similarly distributed across the 3 cohorts, with cluster 3 being the most common CNA cluster (>50% of samples) and cluster 2 being only slightly less common (30-40% of samples). Intriguingly, we did notice that within the TCGA cohort, cluster 1 was more frequent than in the other two cohorts (**Table 1** below). This can be attributed to the fact that the TCGA cohort mainly consists of stage 2 or 3 CRC tumors, whereas in both APD and CAIRO mCRC tumors are much more frequent. It is indeed established that MSI tumors are more frequent in stage 2-3 tumors and because they typically belong to CNA cluster 1, the percentage of CNA cluster 1 tumors is expected to be higher in TCGA. When additionally stratifying the TCGA cohort into metastatic *versus* non-metastatic tumors, frequencies of CNA cluster 1 in metastatic tumors (6.3%) were similar to those in CAIRO, whereas in non-metastatic tumors they were significantly higher (28.3%). Data are added as Supplemental Figure 4d.

Table 1: Distribution of the different cohort in the 3 clusters. The increased contribution of TCGA samples to cluster 1 can be attributed to the primary tumors.

	Cluster 1	Cluster 2	Cluster 3
APD	12.3%	30.9%	56.9%
CAIRO	9.3%	40.5%	50.2%
TCGA	25.3%	37.7%	37.1%

	Cluster 1	Cluster 2	Cluster 3
APD	12.3%	30.9%	56.9%
CAIRO	9.3%	40.5%	50.2%
TCGA (metastatic)	6.3%	37.5%	56.3%
TCGA (stage 1, 2 or 3)	28.3%	38.6%	33.2%

- We also assessed whether each of the cohorts had a similar outcome (overall survival, OS) or were similarly distributed with respect to age, gender, grade and stage. We observed that patient characteristics were largely similar across the 3 cohorts, especially with respect to age and gender. There was a large and significant difference for stage between TCGA, APD and CAIRO (**Figure 1**). This is related to the inclusion criteria from the 3 cohorts that were assembled (TCGA represents unselected CRC and thus has all stages, CAIRO only metastatic CRC, while APD represents metastatic CRC, but $\pm 50\%$ of APD tumor biopsies were collected during resection of a previous CRC, see below question 3 from this reviewer). With respect to grade, data from the TCGA cohort were lacking. However, because in APD tumor biopsies were also collected during a previous surgery (and therefore enriched for grade 2), grade 3 tumors were more frequent in CAIRO than APD. Data are added as Supplemental Figure 1a-e.

Figure 1: Distribution of clusters, gender, grade, age and stage in the 3 different cohorts (APD, CAIRO and TCGA). P-values are respectively $1.49 \cdot 10^{-E7}$ (for CNA cluster distribution), 0.011 (gender), $<10^{-16}$ (grade) and $<10^{-E16}$ (stage) by Chi2-square. Age distribution per cohort was assessed by ANOVA ($P=6.76 \cdot 10^{-E8}$).

When assessing differences between the cohorts at the level of OS, we noticed that (as expected) the TCGA cohort had a better prognosis (**Figure 2**). When stratifying TCGA samples into metastatic versus non-metastatic samples, the TCGA metastatic samples and the CAIRO cohorts behaved similarly as the APD cohort (P=0.378 and P=0.109, respectively). Data are added as Supplemental Figure 1f.

Figure 2: Overall survival analysis using Cox regression for the APD, CAIRO, TCGA metastatic samples (stage 4) and TCGA non-metastatic samples, while correcting for covariates (age, gender and TNM stage). APD was considered the reference cohort.

- To check whether in each of the cohorts the same amplifications and deletions were detected, we re-analyzed CNA data separately for each cohort (TCGA, APD and CAIRO). We observed that the vast majority of peaks detected in the separate cohorts were in common with the peaks of the whole dataset (

Figure 3). However, as could be expected, some CNA regions that were significantly enriched in the whole cohort, were also clearly present in the separate cohorts, but failed to reach statistical significance. This is, however, expected since reduced sample sizes in the separate cohorts also reduces sensitivity of detection. Data are added as Supplemental Figure 3a-b.

Figure 3: (left) The total number of significant GISTIC peaks detected in each cohort (whole cohort, APD, CAIRO and TCGA) is shown in red. For each cohort, we also calculated how many of these peaks were detected in the whole cohort (102 focal peaks) as shown in blue. (right) The proportion of peaks per cohort that are present in the whole cohort is shown (blue divided by red number of peaks in the left plot).

- Finally, we also investigated the average frequency of the 43 focal amplifications and 59 focal deletions per sample and compared this between the 3 cohorts. We observed that in TCGA tumors the frequency was substantially lower than in APD or CAIRO tumors ($P=5.57E^{-08}$ and $P=2.77E^{-13}$, respectively, **Figure 4**). In the next question of the reviewer we have addressed the reason for this reduced frequency in TCGA and show that this is most likely due to the fact that lower stage tumors, which represent a large fraction of TCGA tumors, have fewer CNAs. Data are added as Supplemental Figure 3c-d.

Figure 4: The average number of amplifications (left) or deletions (right) detected per samples across the 3 cohorts (APD, CAIRO and TCGA).

This was also reflected in the distribution of individual CNAs across the 3 cohorts. With the exception of 3 focal events, which were clearly more frequent in the APD cohort, we observed a very stable distribution of CNA events across the cohorts. Specifically, CNAs had a frequency of ~25% in TCGA, and ~35-45% in both APD and CAIRO (

Figure 5). Notably, the 3 focal events with increased frequency in APD did not show specific characteristics. For instance, they were not smaller in size than the other events, were not located on a specific chromosome, etc. Data are added as Supplemental Figure 3e.

Figure 5: Percentage of samples (%) with one of the 102 focal events stratified for the 3 cohorts (APD, green; CAIRO, yellow and TCGA, brown).

2. Despite being considered metastatic tumors, 106 cases from the APD cohort were collected prior to metastasis. Their copy number data really reflect the genome of the primary tumor but not the metastasis stage. It is unclear how these 106 cases are distributed across the 3 clusters, and furthermore, when bevacizumab was administered to these patients compared with tumors collected after metastasis.

The reviewer indeed correctly notes that 80 (and not 106) APD patients had a resection for a colorectal tumour prior to developing a metastatic CRC, for which they were subsequently treated with chemotherapy ± BVZ. For these 80 patients, tumour biopsies were collected at resection for the prior CRC (referred to as CRCp) and not at the time of metastatic disease relapse. We included for these samples the tumour characteristics (grade and stage) at the time of surgery, whereas for the survival analysis we used PFS and OS data obtained at disease relapse, i.e., during treatment with chemotherapy ± BVZ. Age at the time of disease relapse was considered in all survival analyses. This now raises two questions:

1/ Are CNA clusters in APD similar between tumours collected at resection (from patients who later develop a mCRC) compared to tumours collected from patients presenting with primary metastatic disease.

In the revised manuscript, we repeated the CNA cluster analysis on the metastatic CRC (mCRC) samples only. Particularly, this analysis was now performed on 392 mCRC tumours (n=124 from APD, n=205 from CAIRO and n=63 from TCGA). We still identify 3 CNA clusters in the remaining 392 mCRC tumours, and we show how 91.5% of the samples are classified in the same cluster. Of notice, 7.2% of the samples switch between clusters 2 and 3 and only 1.1% of the samples switch from cluster 1 to 2, or *vice versa*. The observed distribution of CNA clusters in the 80 APD tumours collected during resection and 124 mCRC APD tumours was also very similar: CNA cluster 1: 15% and 14%, CNA cluster 2: 31% and 35%, CNA cluster 3: 54% and 51%. In the revised manuscript, we now use this cluster assignment for the 392 mCRC samples.

Table 2: We applied our CNA clustering approach to classify the 402 metastatic CRC biopsies and compared clustering results to those described in our original manuscript. Most tumors samples (>91%) cluster in the same CNA cluster (green shaded cells).

		Metastatic CRC (n=402)		
		Cluster 1	Cluster 2	Cluster 3
All CRC samples	Cluster 1	9.7%	0.7%	0.2%
	Cluster 2	0.2%	35.6%	3.7%
	Cluster 3	0.0%	3.5%	46.3%

The above analyses suggest that CNA clustering based on the 102 CNAs detected by GISTIC in the whole cohort classify the large majority of metastatic tumours in the same CNA cluster. Nevertheless, CNAs of the CRCp APD tumours may still evolve and may have changed upon metastatic disease relapse. This could indeed be the case, because in our original manuscript we described how CNA clusters 2 and 3 contain a higher proportion of stage 4 CRC tumours than CNA cluster 1. It is thus possible that CNA cluster 1 tumours at resection accumulate more CNAs at metastatic relapse and at metastasis would cluster in CNA cluster 2 or 3. We believe, however, that this is not a frequent event. Indeed, there are more stage 4 tumours in CNA cluster 2 or 3 because MSI tumours (which typically cluster in CNA cluster 1) less often progress into mCRC. Indeed, when assessing MSI status in the 80 CRCp APD tumours and comparing against the 124 mCRC APD tumours, we observed no difference in

MSI frequency, i.e., 15% versus 14%. Furthermore, when comparing the number of CNA events between CRCp and mCRC APD tumours, we observed no significant difference ($p>0.5$, Student's *t*-test) between the number of breakpoints or the fraction of the genome with chromosomal instability (CIN) or the overall survival between samples collected before or after metastasis (

Figure 6). Although we agree that in general tumours will gain CNAs when they progress to higher stages, our analyses suggest that this has no major effect on CNA cluster identity. Data are added as Supplemental Figure 7a-e.

Figure 6: Distribution of the 3 CNA clusters in the APD tumours collected at resection ($n=80$, CRCp) and at metastasis ($n=124$, mCRC), as well as the number of breakpoints, the average degree of chromosomal instability (CIN), the number of focal amplifications and focal deletions. For none of the comparisons a significant effect was observed ($P<0.05$).

In the figure below (Figure 7), we also illustrate that limiting ourselves to the 392 metastatic CRC samples only, does not affect the association between CNA clusters and chromosomal instability (breakpoints and proportion of the genome that is unstable), mutation load, the presence of specific CRC driver mutations or grade, T or N-stage. Data are added as Supplemental Figure b-c.

Figure 7: The correlation between the 3 CNA clusters detected in the 392 mCRC samples and the proportion of the genome affected by CNAs, the number of breakpoints and mutations detected, as well as the individual mutation status (for PIK3CA, BRAF, KRAS, APC, TP53), the mutation load, POLD1/POLE mutation status, MSI status, for grade, N-stage and T-stage is shown.

In his/her first question, the reviewer enquired about whether cases from different sources cluster together and whether patient cohorts have similar outcome, age, sex and grade distributions. Therefore, we also characterized CRCp and mCRC APD cohorts in further detail. Briefly, we compared *i*) the whole study population (which included all APD, CAIRO and TCGA samples, N=908), *ii*) 392 metastatic CRC samples only (124 from APD, 205 from CAIRO and 63 from TCGA), and *iii*) 80 CRC patients for which biopsies were taken prior to development of metastatic disease relapse and treatment with BVZ. We specifically, compared CRCp versus mCRC APD samples for tumor characteristics and OS, as shown in the figures below (**Figure 8**). This analysis revealed that although CRCp APD tumors have tumor characteristics from those collected during surgery, their CNA clusters and their survival upon disease relapse do not seem to differ substantially from mCRC in APD (**Figure 9**).

Figure 8: Distribution of clusters, gender, grade, age and stage in the *i*) the whole cohort, *ii*) all 392 mCRC samples (124 mCRC APD, 205 CAIRO and 63 TCGA), and *iii*) 80 CRCp samples. P-values are calculated between the 392 mCRC samples and 80 CRCp samples, are respectively 0.079 (for CNA cluster distribution), 0.459 (gender), $6.44 \cdot 10^{-E7}$ (grade) and $7.21 \cdot 10^{-E14}$ (stage) by Chi2-square. Age distribution per cohort was assessed by ANOVA ($P=0.708$).

Overall survival on the different sample cohorts

Factor	HR	CI-95	P-value
CRCp APD	1.00	(0.72-1.39)	1.0000
CRCm APD	0.83	(0.64-1.08)	0.1704
Age	1.01	(1.00-1.02)	0.0129
Gender	0.84	(0.69-1.02)	0.0773
T-stage	1.38	(1.15-1.65)	<0.001
N-stage	1.22	(1.09-1.38)	<0.001
M-stage	1.08	(0.83-1.42)	0.5558

Progression free survival on the different sample cohorts

Factor	HR	CI-95	P-value
CRCp APD	1.00	(0.73-1.36)	1.0000
CRCm APD	0.78	(0.61-1.00)	0.0456
Age	1.01	(1.00-1.02)	0.1149
Gender	0.94	(0.78-1.12)	0.4724
T-stage	1.32	(1.12-1.55)	<0.001
N-stage	1.10	(0.99-1.22)	0.0872
M-stage	1.32	(1.03-1.70)	0.0291

Figure 9: PFS and OS Kaplan-Meier survival curves for the indicated cohorts. None of the 3 cohorts differ with respect to PFS or OS.

3. In their attempt to replicate the clusters using mCRC cases only, did they re-do GISTIC or did they use the same GISTIC output from the 908 CRCs? If some events are primary tumor specific, it would not make sense to include them in the mCRC analysis.

This question has already been addressed in the answer to the previous question from this reviewer.

4. Most of hypermutators are classified into cluster 1, raising the question whether hypermutation is a relevant factor here. The authors did compare MSI vs MSS tumors in cluster 1 regarding response to bevacizumab, and suggested it was not significant. However, the sample sizes of both groups are very small (MSI, n=11; MSS, n=18). It is likely that the absence of significance is due to the fact that they do not have the statistical power due to the small cohorts. Why not include hypermutation in the multivariate analysis? Or what is the survival differences and response to bevacizumab between cluster 1 and cluster 2&3 if they exclude hypermutators in the comparison?

We thank the reviewer for these valid suggestions and expanded this analysis in the different ways that he/she suggested. We defined hypermutation status as indicated on the figure below (**Figure 10**). Overall, we identified 4 POLDD1 or POLE-mutated tumors, 9 hypermutators and 8 MSI-positive tumors.

Figure 10: Overview of hypermutators distributed over the CNA clusters. Hypermutation definition is based on the number of detected mutations, presence of MSI and POLDD1 and POLE mutations. Hypermutated and non-hypermutated tumors are clearly separated (dotted line).

We then performed the following 2 analyses:

- First, we included hypermutator status in the multivariate analysis as a separate co-variate. We observed that hypermutator status did not have a significant effect on survival outcome: not when comparing survival between the 3 CNA clusters treated with BVZ (**Figure 11**), nor when assessing survival of samples in CNA cluster 1 stratified for BVZ-treatment (**Figure 12**).
- Next, we excluded all hypermutators from the analysis. **Figure 10** gives an overview of the samples that we considered to be a hypermutator and that were removed. Particularly, this

analysis revealed that there is still a significant survival difference between cluster 1 and cluster 2 and 3 for BVZ-treated patients (

- **Figure 13**). Furthermore, within cluster 1, no difference in survival was noted when excluding all hypermutators and stratifying patients for BVZ-treatment (**Figure 14**).

Although we are aware that the sample numbers in CNA cluster 1 are low and this can influence, all additional analyses point to the fact that also non-hypermuted samples that are characterized by a low degree of copy number instability have decreased survival upon treatment with BVZ and that this can not only be attributed to hypermutator status alone. Data are added as Supplemental Figure 11a-d.

Figure 11: Cox regression multivariate analysis testing survival differences between the 3 CNA clusters (while considering CNA cluster 1 as a reference) in BVZ-treated patients. This analysis was corrected for standard covariates, as well as hypermutator status.

Figure 12: Cox regression multi-variate analysis testing BVZ versus no BVZ treatment in CNA cluster 1 patients only, while including hypermutator status as one of the co-variates (in addition to chemotherapy backbone (doublet), age, gender, T-stage).

Figure 13: Cox regression survival analyses on 175 BVZ-treated patients only, comparing the effects between the 3 CNA clusters (considering CNA cluster 1 as a reference), while excluding hypermutator samples as identified in figure 10. The analysis was corrected for age, gender, T and N stage and chemotherapy backbone (doublet).

Figure 14: Cox regression survival analyses comparing BVZ versus non-BVZ-treated patients in CNA cluster 1 patients (considering the non-BVZ treated arm as a reference), while excluding hypermutator samples as identified in figure 10. The analysis was corrected for age, gender, T and N stage and chemotherapy backbone (doublet).

5. The authors failed to provide any insights into the biological differences of the three clusters other than comparing mutation frequencies of a few genes and several histopathological parameters. Excluding hypermutators (MSI plus POLE/POLD1), is there still an enrichment for PIK3CA and BRAF mutations in cluster 1?

After excluding hypermutators (as we defined them in **Figure 10**), we still found a significant enrichment for PIK3CA mutations. Particularly, PIK3CA mutations were more frequent in CNA cluster 1 (33% versus 10 and 0% for CNA cluster 2 and 3, respectively,

Figure 15: The distribution of PIK3CA and BRAF somatic mutations is shown over the 3 CNA clusters in the chemotherapy±BVZ treated patients (N=409). P values calculated by Fischer Z analysis are indicated below each gene.

6. Did they compare the expression of the three clusters using TCGA data?

As requested, we now performed a differential expression analysis on all genes comparing their expression for each individual CNA cluster versus the two other CNA clusters combined. This allowed us to gain biological insights into the 3 CNA clusters. We then applied gene set enrichment analysis with the GSVA bioconductor package [Hänzelmann et al, BMC Bioinformatics 2013]¹. We tested the activity of the 50 hallmark pathways defined by msigdb [Liberzon et al. Cell, 2015]². Next, we calculated the fold-change value for each cluster compared to the other clusters. This revealed a number of pathways that were more strongly activated in each CNA cluster. Particularly, pathways

overrepresented in CNA cluster 1 were those involved in the interferon alpha and gamma response, in allograft rejection, IL6/JAK/STAT3 signalling and complement activation, which are pathways that are all indicative of a tumor microenvironment that is strongly repressing the anti-tumor immune response, consistent with the fact that CNA cluster 1 tumors are hypermutators with high neo-epitope load. Other pathways that were upregulated in CNA cluster 1 tumors included bile acid metabolism, glycolysis, RAS signalling and reactive oxygen species pathways. Finally, CNA 1 clusters were also significantly enriched for the hypoxia pathway. CNA 2 and 3 clusters were characterized by upregulation of similar pathways, although enrichment was more prominent either in cluster 3 (for angiogenesis, myogenesis or epithelial-to-mesenchymal transition), or in cluster 2 (for WNT/beta-catenin signalling, inflammatory response, peroxisome signalling). Pathways that did not differ between the 3 CNA clusters were the MYC, PI3K/AKT/MTOR and TP53 signalling pathways, the DNA repair and G2M checkpoint pathways, as well as the oxidative phosphorylation and fatty acid metabolism pathways. These data are visualized in a heat plot, as shown below (**Figure 16**) and have also been included into the main manuscript figures (see Figure 7c).

Figure 16: Heatmap plot showing the changes in pathway expression for each CNA cluster versus all other CNA clusters.

7. Based on the copy number pattern, it appears what differs between cluster 2 and 3 is DNA loss – cluster 3 has many more recurrent losses than cluster 2. What are these losses, and is there any genes/regions consistently lost in cluster 3 but not 2? Can they provide a IGV screenshot in the supplement so that we can appreciate the copy number landscape of the three groups?

We have now visualized the percentage of samples affected by an amplification or deletion on each of the chromosomes in an IGV plot (**Figure 17**). This was done for all CRC samples (n=908) and separately for samples categorized in CNA cluster 1, 2 and 3. This IGV plot convincingly shows that tumors in CNA cluster 1 are almost devoid of CNAs, with the exception of some whole-arm

amplifications (e.g., affecting chromosome 7, 8, 12 and 13) present in a minority of tumors. On the other hand, cluster 2 and 3 are clearly a lot more frequently affected by CNAs. Several amplifications have the same frequency between cluster 2 and cluster 3 tumors. Particularly, there are several whole-arm amplifications that seem to be present in almost all tumors belonging to cluster 2 and 3 (e.g., chromosomes 7, 8, 13 and 20). Likewise, deletions affecting chromosome 8, 14, 15 and 18 have a more or less similar frequency between both clusters.

Besides these similarities, cluster 2 and 3 also clearly differ in the frequency of the deletions that are detected. Particularly, deletions affecting chromosomes 4, 5, 21 and 22 seem to be significantly more frequent in CNA cluster 3 (**Figure 17**). The same is true for amplifications, affecting chromosome 2, 3, 5 and 6 that are clearly more frequent in CNA cluster 3. To assess whether CNAs were consistently lost in CNA cluster 3 but not in cluster 2, or *vice versa*, we performed GISTIC on tumors from the separate CNA clusters. On **Figure 18**, we show how all 102 peaks identified within the whole cohort are also identified in CNA cluster 2 and 3. On **Figure 19**, we show the frequency of the 102 CNAs in each of the 3 clusters and illustrate how several of them are either as frequent in CNA cluster 2 versus 3, or are more frequent in either CNA cluster 2 or CNA cluster 3. This clearly shows that all 102 peaks are present in CNA cluster 2 and 3, but that there are some differences with respect to their relative frequency in the various clusters. Figure 19 is also included in the main manuscript as Figure 1d.

Figure 17: IGV plot showing how frequent each of the chromosomal regions is affected by amplifications (red) or deletions (blue) in the whole cohort (all samples), in CNA cluster 1, 2 and 3. This plot was included as a main figure in the revised manuscript (see Figure 1b).

Figure 18: (left) The total number of significant GISTIC peaks detected in each CNA cluster is shown in red. For each cluster, we also calculated how many of these were detected in the whole cohort (102 focal peaks) as shown in blue. (right) The proportion of peaks per cohort that are present in the whole cohort is shown (blue divided by red number of peaks on the left plot).

Figure 19: For each of the 102 amplifications or deletions (X-axis) the respective frequency (% of samples affected) in CNA cluster 1, 2 or 3 is shown.

8. In the survival analysis of the validation cohort (Figure 7), why is disease stage is not controlled for as a covariate, similar to the discovery cohort?

We did not correct for T and N stage in the MoMa trials because this study only included metastatic stage IV tumors. Typically, these patients do not undergo resection of their primary tumors and therefore it is not possible to obtain a pathological T or N stage from these patients. On the other hand, we could have obtained clinical staging in these patients, but this would not be very accurate since CT scans or other staging exams that are routinely performed are not able to define T or N for colon cancer. Moreover, we don't really need T and N stage because all of the MoMa patients were diagnosed with metastatic CRC and therefore their staging is IV. Including this variable in the survival analysis would therefore not have made any difference.

Minor criticism

9. Cohort sizes are often inconsistent throughout the manuscript. While the authors initially suggested in "study population" that 274 tumors were collected, table 1 indicates the total number is 215. I assume these 215 cases were those who were subjected to low pass sequencing, but in the table the numbers do not add up to 215 (for instance, 79 females, 134 males, total 213). I guess this might be due to missing info. Nevertheless, the authors need to make a note of it.

The reviewer is correct to notice that we did not indicate missing data in the tables that describe our patient cohort. This has been adapted in the revised manuscript, in Table 1 and Supplementary Table 3. We have moreover carefully assessed whether correct numbers are mentioned in each analysis, table, figure or figure legend.

10. In the section "Clinical and genomic characterization of CNA clusters in mCRC", paragraph 2, the manuscript states that "a multivariable Cox regression to assess prognostic effects of the 3 CNA clusters in mCRC revealed that both CNA cluster 2 and 3 correlated with poor outcome." This is apparently at odds with their data (Figure 2c and 2d), which show the opposite. Even the odds ratio in the next sentence suggests otherwise.

We apologize for the confusion. Both cluster 2 and 3 correlate with 'improved' rather than with 'poor' outcome relative to cluster 1. This has now been corrected in the revised manuscript.

11. In the section "Patients in CNA cluster 2 and 3 benefit from BVZ combination therapy", they write In non-BVZ treated patients (n=224), hazard ratios were not significant, neither for PFS (HR=0.57, P=2.4x10⁻², CI=0.35-0.93 and HR=0.72, P=0.18, CI=0.45-1.16 for cluster 2 and 3 respectively) nor OS (P=0.14, HR=0.66, CI=0.38-1.16 and P=0.53, HR=0.84, CI=0.49-1.44 for cluster 2 and 3 respectively) ...". Why is p=2.4x10⁻² not significant?

In the revised manuscript, we have changed this sentence as follows: "In non-BVZ treated patients (n=224), hazard ratios were not significant for CNA cluster 3 (HR=0.72, P=0.18, CI=0.45-1.16 for PFS and HR=0.84, P=0.53, CI=0.49-1.44 for OS), while for CNA cluster 2 patients, a borderline significant effect was observed for PFS (HR=0.66, P=0.14, CI=0.38-1.16), which was not confirmed at the OS level (HR=0.66, P=0.14, CI=0.38-1.16) ...".

12. I would suggest them to merge the section of comparison with consensus subtype to the section of cluster identification, because I think the flow reads better.

As requested, we have merged both sections in the revised manuscript.

Reviewer #2 (Remarks to the Author):

The manuscript describes quite a lot of background information in a larger data set and then focuses on BVZ-treated patients. Most of these data confirm previous findings, which is fine.

As suggested by reviewer 3, we now shortened the manuscript. We mainly removed some of the background information that the reviewer is referring to. As such, this part of the manuscript has now significantly been shortened.

The BEV sample set, on which the manuscript focuses, actually involves a modest size number of samples (150-200 approximately from Table 1). There is no interaction P value presented for BVZ x CIN in the main analysis and this critical omission makes the major conclusion of the study untenable. (Significant P values in one group but not another are not sufficient here). The interaction P value ($p=0.033$) presented is only after groups 2 and 3 have been re-assigned using a modified definition of high and low CIN using an "optimum" cut-point. This analysis is an important concern.

As requested, we now included an interaction term in several other key analyses in the revised manuscript. Particularly, in the analysis where we assess the effect of BVZ in each of the CNA clusters now also includes an interaction between CNA cluster identity and BVZ. Notably, this interaction was significant both for CNA cluster 2 and cluster 3 (i.e., $P=0.0400$ and $P=0.0108$, respectively). These data have now been added to the revised manuscript.

Further comments:

3. The sample set is very heterogeneous in terms of origins, therapies and follow-up.

The lack of much of the clinicopathological data from the BEV set and of the treatment data used in the wider data set is a concern.

We fully acknowledge the fact that our cohorts are heterogeneous in terms of origin. However, with respect to the wider data set (which we for the sake of clarity here consider to be the TCGA dataset), we would like to stress that we only used this cohort to identify the 3 CNA clusters. Additionally, TCGA was used to link these CNA clusters with gene expression data and characterize the different biological pathways overrepresented in each of the clusters. TCGA data were, however, not used to assess treatment outcome.

In the revised manuscript, we have now much better clarified in a fully transparent matter the origin of the samples. This now makes it much more clear that CRC samples from various origin were used to construct and characterize the CNA clusters. However, when assessing the effect of BVZ samples were less heterogeneous and the large majority of samples (>90%) involved mCRC patients treated with BVZ in the first-line setting. Furthermore, with respect to the heterogeneity of the BVZ set, we would like to stress that our replication cohort (MoMa trials) was a phase 2 clinical trial, which we not only included to serve as a validation cohort, but also because it consists of a more homogenous set of CRC patients that all presented with primary metastatic disease.

In the discussion of the revised manuscript we have now also added a sentence highlighting that the retrospective nature of our cohort as well as potential selection bias in our sample population represent limitations of this study.

4. The inclusion of cancers treated with curative intent and those with metastatic disease at presentation creates problems in terms of the most appropriate measure of outcome to use in the full data set.

The reviewer indeed correctly notes that 80 APD patients had a resection for a CRC prior to developing a metastatic CRC, for which they were subsequently treated with chemotherapy \pm BVZ. For these 80

patients, tumor biopsies were indeed collected at resection for the prior CRC (CRCp) and not at the time of metastatic disease relapse. We included for these samples the tumor characteristics (grade and stage) at the time of surgery, BUT for the survival analysis we used PFS and OS data obtained at disease relapse, i.e., during treatment with chemotherapy \pm BVZ. Age at the time of disease relapse was considered in all survival analyses. Numerous other studies have used a similar approach to obtain a sufficient number of patients treated with BVZ. Furthermore, the measure of outcome appears to be uniform (at the level of PFS and OS) between the 80 APD patients with a resection prior to mCRC and the 124 APD patients presenting with metastatic disease.

Indeed, in Figure 8 and 9 (reviewer 1), we clearly show how both these cohorts (80 CRCp APD patients and 124 mCRC APD patients) differ with respect to tumor characteristics, but not with respect to survival. Also, when assessing the effect of BVZ separately in each of the 3 CNA clusters, we observed that BVZ prolonged survival compared to standard of care chemotherapy both in CRCp and mCRC APD patients that cluster in CNA cluster 2 and 3, but not in CNA cluster 1 (**Figure 20**). We thus observed the same effects as in the whole chemotherapy \pm BVZ cohort. For instance, in CNA cluster 1 HRs were 1.16 and 1.58 ($P=0.757$ and 0.419 , respectively), whereas in CNA cluster 3 the HRs were respectively, 0.66 and 0.67 ($P=0.0388$ and 0.0927). The fact that some of these comparisons were only borderline or not significant most likely relates to the fact the size of both cohorts was smaller ($n=80$ and 134 patients, respectively). These data have been added to the revised manuscript as Supplemental Figure 10.

Figure 20: Cox regression survival analysis assessing the effect of BVZ ± chemotherapy in CRC patients belonging to CNA cluster 1, 2 or 3, respectively. Standard-of-care chemotherapy was used as a reference, and the APD cohort receiving BVZ was stratified into CRCp APD patients (n=80) and mCRC APD patients (n=124). All analyses were corrected for chemotherapy backbone (doublet), age, T and N stage.

5. The minimum 30% tumour cell content required is below the lower limit used in most studies and could easily lead to lack of sensitivity.

We have done a number of calculations to address this remark:

- The 30% tumor cell content that was mentioned in the original manuscript, was estimated by our central pathologist. For the DNA extraction, we performed a macro-dissection on those regions enriched for tumor cells. As a result, tumor percentages from DNA-extracted areas are often higher than 30%. This has now been clarified in the revised manuscript.
- For all samples, on which we performed low-pass whole-genome sequencing and exome-sequencing, we used ASCAT to determine accurate tumor percentages based on B-allele frequencies and LogR of on average $82\,127 \pm 16\,301$ SNPs per sample detected in the exome-sequencing data. We observed that tumor percentages in CNA cluster 1 tumors are slightly lower than in CNA cluster 2 and 3 tumors (Error! Reference source not found.). However, when stratifying these tumors in 3 groups; i.e. those with tumor percentage <40%, with tumor percentage 40-60% and tumor percentage >60%, we detect a similar number of breakpoints and copy number aberrations. Therefore, using a minimum of 30% tumor cell content does not seem to cause a lack of sensitivity.
- Furthermore, it is known that samples from the CMS1 subtype generally have a high number of infiltrating immune cells. This most probably explains why CNA cluster 1 tumors have slightly lower tumor percentage.

Figure 21: Effect of tumor percentage on chromosomal instability. (left) Shown is the average tumor percentage as detected based on exome-sequencing distributed across the 3 CNA clusters. (middle) The number of CNA breakpoints detected in tumors with low (<40%), medium (40-60%) and high (>60%) tumor percentage are depicted. (right) Chromosomal instability (CIN) detected in tumors with a low (<40%) medium (40-60%) and high (>60%) tumor percentage are depicted. These data were added to the manuscript as Supplemental Figure 5.

6. Sequencing quality metrics are lacking.

We apologize that this information was missing and now added the sequencing quality metrics in the supplement. For each sample, we have now added a table highlighting: Sample ID, total Reads, percentage unique reads passing filter aligned, unique bases passing filter aligned, on bait bases, near bait bases, off bait bases, mean bait coverage, percentage usable bases on bait, fold enrichment, percentage zero coverage target, percentage target covered 2x, percentage target covered 10x, percentage target covered 20x, percentage target covered 50x and percentage target covered 100x.

7. Some of the figure legends seem incomplete (e.g. what is SOC?)

We apologize for this oversight and have carefully checked all figure legends in the revised manuscript. SOC refers to standard-of-care chemotherapy, but we have now removed it from all figure legends and panels.

8. It is arguable that the non-BEV arm shows some evidence of an association between good survival and CIN in group 2.

This sentence has now been described as follows: In the revised manuscript, we have changed this sentence as follows: “In non-BVZ treated patients (n=224), hazard ratios were not significant for CNA cluster 3 (HR=0.72, P=0.18, CI=0.45-1.16 for PFS and HR=0.84, P=0.53, CI=0.49-1.44 for OS), while for CNA cluster 2 patients, a borderline significant effect was observed for PFS (HR=0.66, P=0.14, CI=0.38-1.16), which was not confirmed at the OS level (HR=0.66, P=0.14, CI=0.38-1.16) ...”.

9. No estimation of study power for predicting response to BVZ is provided.

As suggested by the reviewer, we now included a power calculation analysis in the revised manuscript. Since it was difficult for us to calculate the study power when assessing 3 groups, which each occur with different frequencies (i.e., the different CNA clusters), we focused this power analysis on comparing the effect of chemotherapy ± BVZ for CNA cluster 2 and 3 tumors combined versus CNA cluster 1. Using the power predictions described by Schoenfeld et al.[Schoenfeld et al.,Biometrics 1983], we set α to 0.05 and ranged β from 1 to 0 (corresponding to a predictive power ranging from 0% to 100%)³. We then calculated the number of events that were needed to reach a specific predictive power using our observed hazard ratio (i.e., HR=0.68) as effect size. This resulted in **Figure 22**, in which the predictive power was plotted against the number of events analysed. Since we analysed n=359 samples of which 331 patients with events, we obtained a predictive power of 94%.

Figure 22: Predictive power of the study to detect a hazard ratio of 0.68 with 95% confidence ($\alpha=0.05$) in function of the number of patients analyzed. The green line indicates the n=331 patients with an event included in this study and the corresponding 94% power of the study.

In a next step, since BVZ- or chemotherapy-treated patients did not contain the same number of samples, we calculated how many events would be required taken into account that both groups were unequally distributed. Considering a ratio of 0.83 between BVZ and chemotherapy-treated patients, we found that the predictive power of our cohort was slightly lower than the 94% we obtained based on Figure 24 (under the assumption of an equal distribution of BVZ and chemotherapy-treated

patients). Specifically, the power of our cohort was found to be 92% (**Figure 23**). This is now clearly highlighted in the revised manuscript on page 12, line 9.

Figure 23: Predictive power of the study to detect a hazard ratio of 0.68 with 95% confidence ($\alpha=0.05$) in function of the number of patients analyzed. The red line indicates the $n=139$ patients with an event included in this study and the corresponding 92% power of the study.

10. Are there other studies on BVZ that have addressed similar questions in colorectal or other cancers? If so, how do they compare? There is no discussion of these.

In the revised manuscript, we have now mentioned the study from Van Dijk et al. which was recently published in JCO⁴. Particularly, in this study a deletion of a specific chromosomal region of chr 18 was found in a candidate gene approach to correlate with BVZ response. In the revised manuscript we have referred to these data as follows (see revised discussion, page 19): “Additionally, our data provide additional insights into the recent observation that chromosome 18q11.2-18q21.1 loss predicts response to BVZ in mCRC. Indeed, our findings suggest that genome-wide instability, rather than the specific loss of one chromosomal region, underlies the association of CNAs with response to BVZ.”

Reviewer #3 (Remarks to the Author):

This article is well written and well designed. It provides some very valuable information on the role of CNAs (Copy Number Alterations) as a potential predictive value for the use of bevacizumab (BVZ) in metastatic colorectal cancer. The authors underline that the benefit of adding BVZ to standard of care combination chemotherapy could be restricted to patients, whose tumors present with higher CNAs. They are able to show that higher quantification of CIN is also related with a more extensive benefit of BVZ treatment. A valuable relation to Consensus Molecular Subtypes (CMS) is also given as well as further verification of the differential response according to CNAs to BVZ in xenograft models.

1. The main limitation of this paper could be the retrospective nature of patient selection for the molecular analysis. To reinforce the validity of their analysis the authors should present this process according to the REMARK criteria. See more details below.

In the revised manuscript, we now present patient selection and inclusion criteria for this study according to the REMARK criteria⁵. We also clearly mention in the introduction that the manuscript follows REMARK criteria. Additionally, we listed the REMARK criteria below and have highlighted where the respective information is included in the revised manuscript.

Introduction

1. State the marker examined, the study objectives, and any prespecified hypotheses.

This has now clearly been stated in the introduction (page 6, last paragraph).

Materials and Methods Patients

2. Describe the characteristics (e.g. disease stage or comorbidities) of the study patients, including their source and inclusion and exclusion criteria. 3. Describe treatments received and how chosen (e.g. randomised or rule-based). See page 21-22 in the Methods section.

Specimen characteristics

4. Describe type of biological material used (including control samples), and methods of preservation and storage. See page 22, methods section.

Assay methods

5. Specify the assay method used and provide (or reference) a detailed protocol, including specific reagents or kits used, quality control procedures, reproducibility assessments, quantitation methods, and scoring and reporting protocols. Specify whether and how assays were performed blinded to the study end point. See page 23-25.

Study design

6. State the method of case selection, including whether prospective or retrospective and whether stratification or matching (e.g. by stage of disease or age) was employed. Specify the time period from which cases were taken, the end of the follow-up period, and the median follow-up time.

See page 21-22.

7. Precisely define all clinical end points examined. See page 22, line 15

8. List all candidate variables initially examined or considered for inclusion in models. See introduction, statistical section.

9. Give rationale for sample size; if the study was designed to detect a specified effect size, give the target power and effect size. A power calculation has now been described in the results section on page 12.

Statistical analysis methods

10. Specify all statistical methods, including details of any variable selection procedures and other model-building issues, how model assumptions were verified, and how missing data were handled.

11. Clarify how marker values were handled in the analyses; if relevant, describe methods used for cut point determination.

An extensive statistical method section is included on page 26 and 27 of the revised manuscript.

Results Data

12. Describe the flow of patients through the study, including the number of patients included in each stage of the analysis (a diagram may be helpful) and reasons for dropout. Specifically, both overall and for each subgroup extensively examined report the numbers of patients and the number of events. A lot of attention has been given to the selection of the patients. We devote almost the first 3 pages of the results to describing how the cohorts were assembled.

13. Report distributions of basic demographic characteristics (at least age and sex), standard (disease-specific) prognostic variables, and tumour marker, including numbers of missing values. See Table 1 and Supplementary Tables.

Analysis and presentation

14. Show the relation of the marker to standard prognostic variables. This has been done extensively for the CNA clusters copy number load, mutations, mutational burden, grade, staging, age, gender, etc.

15. Present univariate analyses showing the relation between the marker and outcome, with the estimated effect (e.g. hazard ratio and survival probability). Preferably provide similar analyses for all other variables being analysed. For the effect of a tumour marker on a time-to-event outcome, a Kaplan–Meier plot is recommended. Kaplan-Meier curves and multivariate Cox regression analyses are provided throughout the manuscript.

16. For key multivariable analyses, report estimated effects (e.g. hazard ratio) with confidence intervals for the marker and, at least for the final model, all other variables in the model. All results from the multivariate analyses are included in the figures, also when these are not significant. Confidence intervals and P-values are standard provided.

17. Among reported results, provide estimated effects with confidence intervals from an analysis in which the marker and standard prognostic variables are included, regardless of their significance. All results from the multivariate analyses are included in the figures, also when these are not significant.

18. If done, report results of further investigations, such as checking assumptions, sensitivity analyses, internal validation. Where appropriate, this has been included.

Discussion

19. Interpret the results in the context of the prespecified hypotheses and other relevant studies; include a discussion of limitations of the study. We have now more carefully discussed the study limitations on page 20 of the discussion.

20. Discuss implications for future research and clinical value. This has been done accordingly. Particularly, we now describe how copy number load can emerge as an additional biomarker next to tumor mutational burden (that measure MSI, POLE and POLD1 mutated tumors).

2. The authors should also make some interpretation on the potential consequences of their findings for future research in the field as well as for clinical practice. See answer to another question from the reviewer below.

See below some recommendations on the specific parts of this manuscript.

The abstract is very clear and delivers a very clean message. Its reading allows a full understanding of the content of the article.

3. In the introduction the authors clearly express the important of the issue they are willing to analyze as well as the main purpose of their research presented in this paper. Reference 1 is outdated (2014) and it should be substituted for a more recent one. We have replaced this reference with a more recent one [Siegel et al., Cancer J. Clin. 2017]⁶.

Results

4. The study population is well described. The method of sample selection is well described. A reference to the REMARK criteria here would reinforce the good methodology used in this article. We have added a reference to the REMARK criteria [McShane et al., British Journal of Cancer 2005] and now also highlight that our description of the patient selection and inclusion follows the REMARK criteria⁵.

5. The section on unsupervised clustering of CNAs does not require any amendment as well as the one on Clinical and Genomic characterization of CNA clusters. However, in the first paragraph of page 11 when they refer to the benefit of bevacizumab in CNA cluster 2 and 3, I wonder if differences in responses (response rate should be given as a proportion or percentage of responding patients) should be better expressed as Odds ratios rather than with Hazard ratios. Check for the same in the second paragraph of page 11 when describing response differences in BVZ treated patients.

We apologize for the confusion. We realize that in the original manuscript, we have sometimes used rather confusing terminology. Particularly, we have referred to 'poor response to BVZ 'when discussing a poor progression-free survival or overall survival in some of the CNA subgroups, rather than describing effects on RECIST response measures. This has now been clarified in the revised manuscript. As such, since we only assessed effects on survival, we still believe that hazard ratios rather than odds ratios are the most appropriate statistical measures to be used.

The sections on the predictive value of CIN for BVZ treatment and its validation in the MoMa trial cohort do not require any modifications. It is likewise for the last two sections on the overlap with CMS clusters and on the use of xenografts.

Tables and figures are informative and well designed. The supplementary material is also of value.

Discussion

It is perhaps too long and it should be shortened. The authors should focus on their findings as well as its potential usefulness and limitations. In that sense, the first paragraph in page 17 could be maintained. However, in the second one, the example of allelic imbalance of chromosome 18 and the one on the value of MET should be removed because it doesn't add any relevance to this publication. I would recommend jumping from "A continuous effort to understand...to disease prognosis³²" to "We therefore explore the prognostic relevance of CNAs...for BVZ have not yet emerged."

As suggested by the reviewer, we have shortened the discussion.

A reference to prenatal diagnosis in the second paragraph of page 19 is not really needed and it should be removed.

This has been changed accordingly.

In page 20 the references to ICON7 and NSABP C-08 are very speculative and not relevant for the discussion of this paper and therefore should be removed. The reference to van Dijk et al on chromosome 19q loss unless already published should also be removed.

This has been changed accordingly. However, we have kept the reference to Van Dijk⁴ because reviewer 2 specifically asked to describe other papers that have looked at CNAs with respect to BVZ response.

In page 21 and 22 the full paragraph dedicated to discuss the potential value of immunotherapy in CNA cluster 1 is somehow out of the main scope of this article and it has to be significantly shortened. I would keep the potential value of associating check-point inhibitors and anti-angiogenics in cluster 1 patients.

We have substantially shortened the discussion, as suggested by the reviewer.

I would agree with the authors that one limitation of this study is the lack of information on tumor sidedness. Not an issue. However, I think is very relevant to discuss on the potential clinical application of the information provided. The retrospective nature plus the potential selection bias in the sample population of this analysis is also limiting. The article states that CIN represents a novel marker for bevacizumab response, but in what sense can it be applied. Do the authors recommend not to use chemotherapy plus bevacizumab in patients classified as CNA cluster 1? What kind of additional information is needed to make a strong statement on this? A comment on these points should be added to the discussion.

We agree with the reviewer and in the revised manuscript we now explicitly describe this as follows: "Recently Le et al. showed how MSI mCRC tumors, which typically are associated with high tumor mutational burden, respond extremely well to PD-1 blockade with pembrolizumab, ultimately leading to the pan-cancer approval of anti-PD-1 therapy for MSI tumors¹⁷. Nowadays, MSI tumors will therefore first receive anti-PD-1 immunotherapy, rather than BVZ combined with chemotherapy. Our data, which show that tumors characterized by low copy number burden do not benefit from BVZ, thus seem to confirm that anti-PD-1 therapy is a better treatment option for these patients. Furthermore, our data suggest that other CNA cluster 1 tumors that are not MSI, also do not benefit from BVZ therapy and might therefore be better of treated with anti-PD-1 therapy. Although this needs to be confirmed in follow-up prospective clinical studies, the use of copy number load as an additional biomarker next to tumor mutational burden, might become clinically useful."

We also added 'potential selection bias in the sample population' and the 'retrospective nature of the population' as limitations of the study (page 20 of the discussion).

References

1. Hänzelmann, S., Castelo, R. & Guinney, J. GSVA: gene set variation analysis for microarray and RNA-Seq data. *BMC Bioinformatics* **14**, 7 (2013).
2. Liberzon, A. *et al.* The Molecular Signatures Database Hallmark Gene Set Collection. *Cell Syst.* **1**, 417–425 (2015).
3. Schoenfeld, D. A. Sample-Size Formula for the Proportional-Hazards Regression Model. *Biometrics* **39**, 499–503 (1983).
4. van Dijk, E. *et al.* Loss of chromosome 18q11.2-q12.1 is predictive for survival in metastatic colorectal cancer patients treated with bevacizumab. *J. Clin. Oncol.* **36**, (2018).
5. McShane, L. M. *et al.* REporting recommendations for tumour MARKer prognostic studies (REMARK). *Br. J. Cancer* **93**, 387–91 (2005).
6. Siegel, R. L. *et al.* Colorectal cancer statistics, 2017. *CA. Cancer J. Clin.* **67**, 177–193 (2017).

REVIEWERS' COMMENTS:

Reviewer #1 (Remarks to the Author):

The authors have addressed all my comments.

Reviewer #2 (Remarks to the Author):

No further comments

Reviewer #3 (Remarks to the Author):

I do thank the authors for performing such an extensive revision of the original manuscript. The addition of the REMARK criteria is very much appropriate. The discussion has been also shortened as recommended, and it is now better oriented. The paper reads now much better than in the original version.